



# Modelling subglacial fluvial sediment transport with a graph-based model, GraphSSeT

Alan R.A. Aitken[1,2], Ian Delaney[3], Guillaume Pirot[1,4], and Mauro A. Werder[5,6]

[1]School of Earth Sciences, The University of Western Australia, Perth Western Australia, Australia
[2]Australian Center of excellence for Antarctic Science , The University of Western Australia, Perth Western Australia, Australia
[3]Institut des dynamiques de la surface terrestre (IDyST), Université de Lausanne, Lausanne, Switzerland
[4]Mineral Exploration Cooperative Research Centre, Centre for Exploration Targeting, School of Earth Sciences, The University of Western Australia, Perth, Australia
[5]Swiss Federal Institute for Forest, Snow and Landscape Research (WSL), Birmensdorf, Switzerland
[6]Laboratory of Hydraulics, Hydrology and Glaciology (VAW), ETH Zurich, Zurich, Switzerland

**Correspondence:** Alan Aitken (alan.aitken@uwa.edu.au)

**Abstract.** A quantitative understanding of how sediment discharge from subglacial fluvial systems varies in response to glacio-hydrological conditions is essential for understanding marine systems around Greenland and Antarctica and for interpreting sedimentary records of cryosphere evolution. Here we develop a graph based approach, GraphSSeT, to model subglacial fluvial sedimentary transport using subglacial hydrology model outputs as forcing. GraphSSeT includes glacial erosion of bedrock and a dynamic sediment model with exchange between the active transport system and a basal sediment layer. Sediment transport considers transport-limited and supply-limited regimes and includes stochastically-evolving grain size, network scale flow management and tracking of detrital provenance. GraphSSeT satisfies volume balance and sediment velocity and transport capacity constraints on flow. GraphSSeT is demonstrated for synthetic scenarios that probe the impact of variations in hydrological, geological and glaciological characteristics on sediment transport over multi-diurnal to seasonal timeframes. For steady-state hydrology scenarios on seasonal timescales we find a primary control from the scale and organisation of the channelised hydrological flow network. The development of grain size dependant selective transport is identified as the major secondary control. Non-steady-state hydrology is tested on multi-diurnal timescales, for which sediment discharge scales with peak water input leading to increased sediment discharge compared to steady state. With increasing application of subglacial hydrology models, GraphSSeT extends this capacity to define quantitatively the volume, grain size distribution and detrital characteristics of sediment discharge, and enables a stronger connection of models of the glacio-hydrological system with constraints from the sediment record and impacts on marine systems.

## 1 Introduction

Discharge of sediments from subglacial fluvial systems is an important component of the ocean system and impacts on the delivery of nutrients (Wadham et al., 2013; Meire et al., 2017; Cape et al., 2019; Overeem et al., 2017), ice-shelf cavity processes (Smith et al., 2019) and marine geomorphology (Dowdeswell et al., 2016, 2015). Turbid sediment plumes provide





a means to observe subglacial fluvial inputs into the ocean that are otherwise unobservable (Schild et al., 2017; Chu et al., 2009). Furthermore, marine sediments from currently or formerly glaciated margins are a primary record of past cryosphere change, and are crucial to establishing pinning points in global climate evolution (Lepp et al., 2022; Hogan et al., 2020, 2011; Witus et al., 2014; Hogan et al., 2012; Andresen et al., 2024). Subglacial fluvial systems comprise a long-term evolution

with superimposed events of short duration but with high flow (Dow, 2022; Dow et al., 2022; Chu, 2014; Ashmore and Bingham, 2014). These are associated with seasonal to diurnal surface melting cycles (Ehrenfeucht et al., 2023) and flood events, including jökulhlaups (Roberts, 2005). Ice stream 'water piracy', involving the re-routing of subglacial water from one ice stream to another, may occur in response to relatively small changes in glacial mass-balance or bed conditions (McCormack et al., 2023; Alley et al., 1994; Vaughan et al., 2008). It is likely that ongoing changes in cryosphere systems will increase the

frequency and intensity of high-flow events in the future, and so increased sediment flux to the ocean (Delaney and Adhikari, 2020)

To express the consequences of cryosphere change on marine systems, and to fully comprehend the implications of sedimentary records of the past and present cryosphere there is a need for a quantitative understanding of subglacial fluvial sediment transport systems (Delaney and Adhikari, 2020). Here we describe a new graph-based approach, GraphSSeT, with capacity

to model subglacial fluvial sediment transport for glacio-hydrological models across a range of time- and length-scales. The approach is demonstrated for subglacial hydrology model outputs from synthetic scenarios (De Fleurian et al., 2018) for which we assess the most important key hydrological, geological, and glaciological controls on the sediment transport system.

## 2   Background

### 2.1   Conceptual Background

### 2.1.1   Subglacial hydrology

Under glaciers and ice sheets water transport occurs in response to hydraulic potential gradients that drive flow towards lower potential areas (Shreve, 1972). The subglacial hydrology system involves numerous components, but fundamentally includes input, storage, flow and output (Ashmore and Bingham, 2014). Water enters the subglacial hydrology system by the melting of basal ice, groundwater discharge, subglacial lake discharge, and surface melt transported to the bed through moulins. In the

context of the modeling to come we make a distinction between a distributed input, occurring over a substantial area relative to the model resolution, and focused input(s) with a high volume input at a specific location(s). Water may be stored englacially and in subglacial lakes, which may be 'active' lakes that fill and empty in a cyclical to episodic fashion, or stable lakes, which indicate balanced flow or closed systems (Livingstone et al., 2022). Water may also be stored in sediments and sedimentary rocks (Christoffersen et al., 2010; Flowers and Clarke, 2002).

Subglacial water flow may be characterised by a distributed, or so-called 'sheet flow', along the bed interface, which is relatively inefficient, and flow concentrated into subglacial channels, so-called 'channelised flow' which is relatively efficient. Distributed flow may be accommodated by several mechanisms including linked-cavity systems (Kamb, 1987), distributed





'canals' eroded into underlying sediment (Walder and Fowler, 1994), thin patchy films (Alley, 1989; Creyts and Schoof, 2009), flow through permeable basal sediments (Boulton et al., 2007); more generally the bed may be viewed as a permeable

interface (Hewitt, 2011; Flowers et al., 2004). Channels may be incised into the overlying ice, forming so-called Röthlisberger or 'R' channels (Röthlisberger, 1972) that evolve dynamically in response to water and ice flow, or alternatively by incision into the bed forming so-called Nye or 'N' channels (Nye, 1976). Outputs from the subglacial hydrology system include discharge to the ocean, re-freezing to the bed (Alley et al., 1998), filling of lakes and englacial reservoirs, or recharge of groundwater aquifers.

The hydrology system is strongly influenced by quasi-stable boundary conditions including bed topography, bed roughness, subglacial geology and geothermal heat flux, but also is intrinsically linked to dynamic changes to the ice that can occur much more rapidly. These including changes in ice thickness and surface slope as well as the thermal state and flow-dynamics of the ice. Particularly important are temporal and spatial variations in hydrological inputs that may occur over timescales from hours to millenia.

**2.1.2 Subglacial sediment transport**

Glaciers can transport sediment coupled with ice flow including as sediment entrained within the basal ice (Licht and Hemming, 2017) and by basal sliding occurring within subglacial till, causing the material above the shear interface to move with the ice (Evans et al., 2006; Christoffersen et al., 2010). These transport processes have been represented in ice sheet models (Pollard and DeConto, 2020) and, although undoubtedly important, are not the focus here. Subglacial fluvial transport is the other major

component of sediment discharge at glaciated margins (Overeem et al., 2017; Andresen et al., 2024), and is more sensitive to changes in glacial-hydrologic state. This highly dynamic spatio-temporal transport system is the focus of our model. Important considerations include the dual limits on transport by transport capacity and by sediment supply, and the interaction of changing hydrology with an evolving basal interface.

The dynamics of subglacial fluvial sediment transport are not well constrained by observations, however, share similarities

with fluvial transport in rivers for which the governing processes are better established (Rijn and C, 2007a, b; Ancey, 2020a, b). Although the physical description of fluvial transport processes is often uncertain with respect to satisfying empirical data, several classes of transport law are well established. One major family of such laws uses the shear stress at the bed to establish transport capacity, using either stochastic or deterministic criteria for sediment mobilisation (Engelund and Hansen, 1967; Meyer-Peter and Müller, 1948; Einstein, 1950). These laws have been applied to model subglacial fluvial sediment transport

in the SUGSET model, with formulations in one-dimension (Delaney et al., 2019) and two dimensions (Delaney et al., 2023). Network-based models are developed for riverine fluvial systems (Czuba, 2018; Wilkinson et al., 2006) but no network-based formulation exists for subglacial fluvial systems.



## 2.2 Modelling Background

### 2.2.1 Subglacial hydrology models

Subglacial hydrology may be described by models with varying degrees of complexity and correspondingly varying computational cost and ease of use (Flowers, 2015). The simplest models use the so-called Shreve potential (Shreve, 1972) to define the direction and relative magnitude of flow (equation 1).

$$\varphi = \rho_i g z_s + (\rho_w - \rho_i) g z_b + N \tag{1}$$

where $\varphi$ is the hydraulic potential at the bed for a given ice-surface $z_s$ and bed elevation $z_b$. $\rho_i$ and $\rho_w$ are the densities of ice
and water respectively, $g$ is gravitational acceleration. $N$ is effective pressure, being defined as ice overburden pressure minus the basal water pressure. Assuming constant $N$, the hydraulic potential gradient may be defined as

$$\nabla \varphi = \rho_i g \nabla z_s + (\rho_w - \rho_i) g \nabla z_b \tag{2}$$

With the inclusion of basal water input, flow can be accumulated to define regions of enhanced flow. For example, Le Brocq et al. (2009) model the spatially variable drainage system as a sheet, the thickness of which is varied to accommodate potential-
gradient driven laminar flow. Shreve potential approaches have the benefit of low computational cost, simplicity, and ease of application to large-scale examples. However, with the assumption of constant $N$, they do not represent well the dynamic interactions of distributed and channelised flow on short timescales nor on length scales of individual channels (De Fleurian et al., 2018).

One more advanced formulation is a network-based approach that considers flow to represent both channels and linked cav-
ities in a unified form as conduits: conduits that exceed a critical discharge represent channels and are spontaneously enhanced due to instabilities between opening and closure rate (Schoof, 2010); the remainder represents the distributed flow system. Other approaches develop a coupled model of the channelised and distributed flow regimes, considering the development of each and the transfer of water between them. Several such models have been developed including descriptions of distributed flow with a so-called 'macroporous sheet' for which thickness varies in-line with effective pressure (Flowers et al., 2004), or
by a linked cavity drainage system (Hewitt, 2011). In the latter, cavity height varies dynamically with an opening rate typically defined from ice-sliding rate and bed roughness, and a closure rate typically defined by ice creep (Hewitt, 2011). R-channels are most commonly represented as circle segments, classically as semi-circles (Röthlisberger, 1972), but also as broad-low conduits with a geometry defined by the so-called Hooke angle (Hooke et al., 1990). The channel sectional area evolves dynamically with a closing rate typically defined by ice creep, and an opening rate typically defined by ice melt at the channel
walls. For our new approach (Fig. 1), any model for which channelised flow can be described on a set of connected edges could be used as input, but perhaps the most widely used approach in recent times, and the one used as forcing here, is the Finite Element Method (FEM) model GlaDS (Werder et al., 2013).



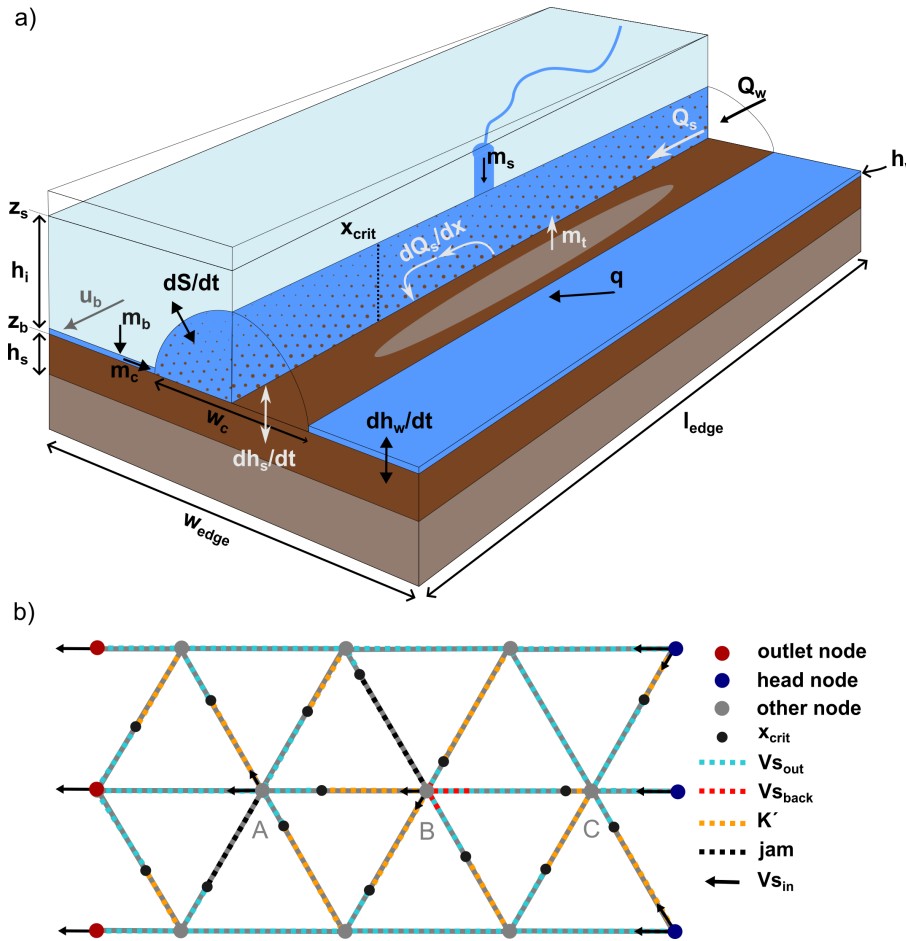

**Figure 1.** a) schematic of a GraphSSeT model edge showing its structure (open arrows) and key model components. Hydrology model components in black, ice sheet model components in grey, sediment model components in white. Variables and quantities for GlaDS and SUGSET are as in sections 2.2.2 and 2.2.3 respectively. b) schematic illustrating the network transport model (see Section 2.2.4). This example shows three interconnected channels each with flow from the head node to the outlet node. The 'northern' and 'southern' channels have supply-limited conditions, where all sediment is able to leave the edge ($V_{s_{out}}$ is sediment leaving the edge). The central channel and connecting edges have transport-capacity limited conditions, and sediment will only leave the edge from below the critical point $x_{crit}$. Furthermore, the edges downstream from node B cannot transport the sediment supplied from upstream. The consequence is for residual sediment volume (as flux density, $K'$) to accumulate on the downstream edges, and ultimately for reduced flow ($V_{s_{in}}$) into the downstream edges, balanced by a back-flow ($V_{s_{back}}$) on the upstream edges: an excessive flux density causes an edge to 'jam'. Jammed edges cannot receive incoming sediment volume, but sediment can flow out to the downstream edges allowing the jam to clear.



### 2.2.2 The GlaDS model

For sheet flow we begin with the conservation of mass

$$\frac{\partial h_w}{\partial t} + \nabla \cdot \boldsymbol{q} = m_b \tag{3}$$

where $h_w$ represents the water sheet thickness, and $m_b$ describes water input to the sheet. $\boldsymbol{q}$ is the sheet discharge and is defined as a Darcy-Weisbach turbulent flow parameterization $\boldsymbol{q} = -k h_w^\alpha |\nabla \varphi|^{\beta-2} \nabla \varphi$, where $k$, $\alpha$ and $\beta$ are defined appropriately for the flow law (Werder et al., 2013). The evolution of the sheet thickness is modelled on elements and is formulated using a linked cavity approach (Hewitt, 2011)

$$\frac{\partial h_w}{\partial t} = \omega - \upsilon \tag{4}$$

where $\omega$ represents the cavity opening rate as

$$\omega(h_w) = \begin{cases} u_b(h_r - h_w)/l_r & \text{if } h_w < h_r \\ 0 & \text{otherwise} \end{cases} \tag{5}$$

with $u_b$ the basal sliding velocity of ice, $h_r$ and $l_r$ are the typical bedrock bump height and horizontal cavity spacing respectively. In equation 4, $\upsilon$ represents the cavity closing rate as

$$\upsilon(h_w, N) = \tilde{A} h_w |N|^{\eta-1} N \tag{6}$$

where $\tilde{A}$ is the ice flow constant scaled for cavities and $\eta$ is the Glen's flow law exponent, typically 3.

Channelised flow is modelled on element edges, with each edge representing a potential channel that can open and close according to the balance between ice creep and melting of the channel walls. Again we begin with the conservation of mass

$$\frac{\partial S}{\partial t} + \frac{\partial Q_w}{\partial x} = \frac{\Xi - \Pi}{\rho_w L} + m_c \tag{7}$$

where $x$ is the coordinate along the edge, $\Xi$ the potential energy dissipated per unit length and time and $\Pi$ the sensible heat exchange of water due to melting or refreezing (see Werder et al. (2013) for definition of $\Xi$ and $\Pi$). $L$ is the latent heat of fusion. $Q_w$ is defined as a Darcy-Weisbach turbulent flow parameterization $Q_w = -k_c S^{\alpha_c} |\frac{\partial \varphi}{\partial x}|^{\beta_c-2} \frac{\partial \varphi}{\partial x}$, with $k_c$, $\alpha_c$ and $\beta_c$ not necessarily the same as those for sheet flow. $m_c$ is the water entering the channel from the adjacent sheet. The time evolution of channel area $S$ is given by the balance of opening rate and closing rate

$$\frac{\partial S}{\partial t} = \frac{\Xi - \Pi}{\rho_i L} - \upsilon_c \tag{8}$$

The closing rate $\upsilon_c$ is defined similarly to eq. 6 replacing $h_w$ with $S$, and with potentially a different scaling for $\tilde{A}$.

The sheet and the channel model are coupled by requiring that the pressure is continuous, i.e. the pressure in the sheet and an adjacent channel are equal. The continuity is assured by fixing the water exchange between the sheet and the channels accordingly (via $m_c$ for the channels and via boundary conditions for the sheet).



### 2.2.3 Subglacial sediment dynamics model

For sediment transport in GraphSSeT, we consider the one-dimensional form of the SUGSET model (Delaney et al., 2019) to apply to individual edges (Figure 1). SUGSET uses an R-channel geometry for which the dimensions of the channel and the channelised flux may be calculated from water inputs. Alternatively, the channel dimensions and flux may be obtained a-priori from an input hydrology model, which is the approach we use here. With knowledge of the channel sectional area and channelised water flux, the basal shear stress from water flow is

$$\tau = \frac{1}{8} f_r \rho_w u_w^2 \tag{9}$$

where $u_w$ is the mean water velocity defined from $Q_w/S$ and $f_r$ the Darcy-Weisbach friction factor. Once $\tau$ is defined the sediment transport capacity may be determined using one of several sediment transport laws. In this case we use the formulation for total sediment flux of Engelund and Hansen (1967).

$$Q_{sc} = \frac{0.4}{f_r} \frac{1}{d_{50} \left(\frac{\rho_s}{\rho_w} - 1\right)^2 g^2} \left(\frac{\tau}{\rho_w}\right)^{\frac{5}{2}} w_c \tag{10}$$

where $d_{50}$ is the median sediment grain size and $\rho_s$ the sediment grain density. $w_c$ is the width of the channel floor, defined for a given $S$ by the Hooke angle of the channel (Delaney et al., 2019; Hooke et al., 1990). Here we use $\pi$, consistent with the R-channel geometry in GlaDS.

In the SUGSET model sediment may be in active transport, or stored in a basal sediment layer, from which sediment can be remobilised. For a channel segment, the sediment in active transport may be limited either by transport capacity (eq. 10) or by a lack of available sediment. SUGSET applies a dynamic bed-evolution that accommodates both transport-capacity-limited and supply-limited regimes with provision for a) the generation of new sediment due to glacial erosion of bedrock, b) mobilisation of existing sediment from the basal sediment layer and c) deposition of excess sediment to the basal sediment layer. In GraphSSeT, glacial erosion potential $\dot{e}$ may be defined either as a power-law of the sliding velocity at the bed (Herman et al., 2018), or, if the basal shear stress of the ice is known, as a linear function of the work done by basal sliding (Pollard and DeConto, 2003):

$$\dot{e} = \kappa u_b^\gamma \text{ or } \dot{e} = \varkappa \tau_b u_b \tag{11}$$

where $\kappa$ is generally between $1 \times 10^{-4}$ and $1 \times 10^{-7}$ m$^{1-\gamma}$/a$^{1-\gamma}$ for $\gamma$ between 1 and 2 (Herman et al., 2018); scaling factor $\varkappa$ may be defined $\approx 2^{-10}$ Pa m/s (Pollard and DeConto, 2003; Golledge and Levy, 2011). Importantly, new sediment generation is restricted by the capacity of the ice to access the bedrock through the basal sediment layer and the following formulation is used to define the source term:

$$m_t = \dot{e} \left(1 - \frac{h_s}{h_{max}}\right) w_{edge} \tag{12}$$

where $h_{max}$ is a limit on the depth to which glacial erosion may penetrate through sediment, typically considered between 0.5 to 2 m (Pollard and DeConto, 2003; Golledge and Levy, 2011; Delaney et al., 2019). Here we use 0.75 m. $w_{edge}$ is the width





of the bed accessible by the channel segment. For this we set $w_{edge} \times l_{edge}$ as one-third the area of an equilateral triangle with side length $l_{edge}$. The mobilisation of sediment from the ice sheet bed is defined as

$$\frac{\partial Q_s}{\partial x} = \begin{cases} \frac{Q_{sc}-Q_s}{l} & \text{if} & \frac{Q_{sc}-Q_s}{l} \leq m_t & \text{(a)} \\ 0 & \text{if} & h_s \geq h_{lim} \ \& \ \frac{Q_{sc}-Q_s}{l} < 0 & \text{(b)} \\ \frac{Q_{sc}-Q_s}{l}\sigma(h_s) + m_t(1-\sigma(h_s)) & \text{otherwise} & & \text{(c)} \end{cases} \tag{13}$$

In equation 13 the first case describes the mobilisation of sediment according using a sediment-uptake e-folding length $l$, here taken equal to $l_{edge}$. If transport capacity ($Q_{sc}$) exceeds the influx of sediment from upstream ($Q_s$) mobilisation will be

positive and sediment will enter the transport system; in the opposing situation, mobilisation is negative and sediment will be deposited to the basal sediment layer. In the first case mobilisation to transport capacity is realised where $(Q_{sc} - Q_s)/l$ is less than the capacity for erosion to supply sediment (equation 13 a). In the second case, we seek to deposit sediment, but already the basal sediment layer has reached the maximum permitted thickness $h_{lim}$ and mobilisation is set to 0 (equation 13 (b)). This prevents 'runaway' accumulations of sediment from occurring in areas such as overdeepenings, as the model has no feedback

between sediment deposition and hydraulics (Creyts et al., 2013). Equation 13 (c) describes a more general case where existing sediment may be mobilised to or from the bed and new sediment may be derived through glacial erosion, depending on the function $\sigma(h_s)$.

$$\sigma(h_s) = \left(1 + \exp\left(10 - 5\frac{h_s}{\Delta\sigma}\right)\right)^{-1} \tag{14}$$

where $\Delta\sigma$ provides a smooth transition between the two terms over the range $h_s = 2\Delta\sigma \pm \Delta\sigma$; for discussion of the influence

of $\Delta\sigma$ see Delaney et al. (2019, 2023). Finally, from the net sediment mobilisation the change in the thickness of the basal sediment layer over time is calculated using the Exner equation

$$\frac{\partial h_s}{\partial t}w_{edge} = -\frac{\partial Q_s}{\partial x}\frac{1}{1-\lambda} + m_t \tag{15}$$

In GraphSSeT we consider porosity as part of the Exner equation with $\lambda = 0.3$ used for the runs performed here. In GraphSSeT, as well as the conditions in eq. 13, sediment mobilisation $\frac{\partial Q_s}{\partial x}$ is further limited as part of the network transport model such

that the condition $0 \leq h_s \leq h_{lim}$ is maintained.

### 2.2.4 Network transport model

The preceding describes the local co-evolution of the basal sediment layer and active transport. In GraphSSeT these are calculated discretely for each graph edge. At network-scale we construct the transport model on the basis of an evolving flux density defined from the sediment volume in active transport on the edge:

$$K = \frac{V_s}{l_{edge}} \tag{16}$$

A limit on flux density that we cannot exceed is defined from the sediment transport capacity as $K_{max} = Q_{sc}dt/l_{edge}$, where $dt$ is the model-time elapsed since the last timestep for that edge, which need not be constant. For each timestep for each edge



we define $V_s$ considering the flow of sediment through the network.

$$V_s = K'l_{edge} + Q_s dt + \frac{\partial Q_s}{\partial x} l_{edge} dt \qquad (17)$$

where $K'$ is the residual flux density from the end of the last timestep on that edge, $\frac{\partial Q_s}{\partial x}$ is the mobilisation of sediment on the edge (eq. 13) and $Q_s$ is the influx to the edge from its upstream node.

In GraphSSeT we control the network-scale flow using three founding principles:

1. Sediment transport can not involve excessive velocities

2. Sediment transport capacity can not be exceeded

3. Sediment volume must be conserved except at the outlets

The first criterion is important in situations where relatively coarse grains are transported on relatively long edges, in which case not all the active sediment is able to leave the edge. To constrain this, we apply an unsteady virtual velocity limit for bedload transport $u_s$ using the empirically-defined relationships of Klösch and Habersack (2018). Empirical relationships with grain size are defined for formula types $u_s = a\left(\tau^* - \tau_c^*\right)\left(\sqrt{\tau^*} - \sqrt{\tau_c^*}\right)$ and $u_s = a\left(\tau^* - \tau_c^*\right)^{\frac{3}{2}}$. Each of these is available in GraphSSeT. In the model runs presented in this study we use the former realised as:

$$u_s = a\sqrt{\frac{\rho_s - \rho_w}{\rho_w} gd_{50}}\left(\tau^* - \tau_c^*\right)\left(\sqrt{\tau^*} - \sqrt{\tau_c^*}\right), \tau^* > \tau_c^* \qquad (18)$$

where $a$ is a dimensionless coefficient, for which Klösch and Habersack (2018) derive the value 2.30. $\tau^*$ is the dimensionless Shields stress $\frac{\tau}{(\rho_s - \rho_w)gd_{50}}$ and $\tau_c^*$ the dimensionless critical Shields stress $\tau_c^* = b\left(\frac{d_{50}}{d_{mean}}\right)^c$ with $b$ and $c$ empirically constrained as 0.052 and -0.82 respectively. $d_{mean}$ is the mean grain size of the population distribution input to the model. For the datasets in Klösch and Habersack (2018) both this formulation and the alternative formulation generated similar results.

From $u_s$ we define the critical point $x_{crit}$ on the edge (Fig.1) as the point below which the sediment is able to reach the downstream node in timestep length $dt$. All active sediment below $x_{crit}$ is assigned to leave the edge in the current timestep. A lower limit on $x_{crit}$, $l_{min}$, is set to avoid stagnant edges, here it is set at $0.1 \times l_{edge}$.

$$x_{crit} = \begin{cases} l_{edge} & \text{if} & u_s dt > l_{edge} \\ u_s dt & \text{if} & l_{min} \leq u_s dt \leq l_{edge} \\ l_{min} & \text{if} & u_s dt < l_{min} \end{cases} \qquad (19)$$

For each edge we define the sediment volume flowing out as $V_{s_{out}} = V_s \frac{x_{crit}}{l_{edge}}$.

The second criterion is important for both controlling the active sediment on the edge and the interactions at nodes where several edges meet. The situation may occur where the maximum flux density for an edge is exceeded by sediment influx from upstream. The edge cannot receive more sediment without violating condition 2, and is designated as 'jammed' (Fig. 1) meaning that neither sediment influx from upstream nodes nor the mobilisation of sediment are permitted. A jammed status does not imply a physical restriction to sediment flow, and outflux to downstream nodes can occur as well as deposition of sediment to the basal sediment layer; therefore the jammed status is inherently transient.



The third criterion is considered at nodes, where we must take the influx from several upstream edges and distribute it between several downstream edges. No volume is stored on nodes, and so to manage the flow distribution we use a kinematic wave approach (Newell, 1993). Total influx to the node may exceed the combined transport capacity of the downstream edges, in which case a 'back-flow' is defined for each node as

$$
v_{s_{back}} = \begin{cases} v_{s_{node}} - v_{sc_{out}} & \text{if} \quad v_{s_{node}} - v_{sc_{out}} > 0 \\ 0 & \text{if} \quad v_{s_{node}} - v_{sc_{out}} \leq 0 \end{cases} \tag{20}
$$

and the corresponding 'out-flow' from the node is defined as $v_{s_{out}} = v_{s_{node}} - v_{s_{back}}$

Transient volume components defined on the nodes include the total incoming sediment volume from upstream edges $v_{s_{node}}$, the combined transport capacity of downstream edges $v_{sc_{out}}$, the total back flow to upstream edges $v_{s_{back}}$ and the resulting out-flow to downstream edges $v_{s_{out}}$. While the back-flow is unphysical, the desired outcome is to limit downstream transport where it is not permissible under criterion 2, while preserving volume balance under criterion 3. The distribution of outflow $v_{s_{out}}$ between the downstream edges is proportional to each edge's contribution to volumetric sediment transport capacity $v_{sc_{out}}$, defined as

$$
V_{s_{in}} = v_{s_{out}} \frac{V_{sc}}{v_{sc_{out}}} \tag{21}
$$

where $V_{sc} = Q_{sc}dt$ for the edge. Similarly, the back-flow $v_{s_{back}}$ is distributed between upstream edges proportional to their contributions to total volumetric sediment input to the node.

$$
V_{s_{back}} = v_{s_{back}} \frac{V_{s_{out}}}{v_{s_{node}}} \tag{22}
$$

From these volumes, the flux density $K = (V_s - V_{s_{out}} + V_{s_{back}})/l_{edge}$ and the incoming flux to the edge $Q_s = V_{s_{in}}/dt$ are updated, ready for the next timestep in which that edge is analysed.

### 2.2.5 Grain size distribution tracking

The grain size distribution is a critical factor for the sediment transport system and is also a key observable for sedimentary records from the ocean and in the subglacial environment. We consider grain size to be a stochastic variable defined by samples from a log-normal population distribution. Studies of till exposures suggest a typical central value on Krumbein's $\phi$ scale of $\approx 2.2$ with standard deviation between 1 and 2 $\phi$ (Peterson et al., 2018; Haldorsen, 1981), so defining the population distribution. This stochastic formulation allows spatial and temporal variability of grain size to be accommodated in the model, but also generates variations in sediment transport capacity and virtual velocity, causing time-variable transport conditions.

In the event that new sediment is required, i.e. at the beginning of the model run and due to erosion, we draw a sample from the population distribution and define for that sample a log-normal sample distribution as

$$
\ln(d) \sim \mathcal{N}(\mu, \varsigma^2) \tag{23}
$$



where $\mu$ and $\varsigma$ are the mean and standard deviation of the sample.

At several times in the transport model it is necessary to mix sediment volumes, for example, when newly mobilised sediment is added to existing sediment on the edge, their respective grain size distributions must be combined. For simplicity, the mixture is defined as a new sample composed from the union of volume-weighted samples drawn from each component distribution:

$$\mathbb{R}_{mixture} = \mathbb{R}_{residual} \cup \mathbb{R}_{influx} \cup \mathbb{R}_{mobilised} \cup \mathbb{R}_{bedrock} \tag{24}$$

with

$$\mathbb{R}_{residual} \sim e^{\mathcal{N}_{edge}}, \mathbb{R}_{influx} \sim e^{\mathcal{N}_{node}}, \mathbb{R}_{mobilised} \sim e^{\mathcal{N}_{sediment}}, \mathbb{R}_{bedrock} \sim e^{\mathcal{N}} \tag{25}$$

Each component is defined by a sub-sample drawn from the sample distribution for that component, with a size proportional to the relative component volume, i.e. for a desired sample size $n$, the sub-sample for a component that provides half the volume has size $n/2$, or the closest integer value. The sub-sample for sediment coming from upstream $\mathbb{R}_{influx}$ is drawn from the distribution for the upstream node $e^{\mathcal{N}_{node}}$, this in turn being defined by the union of volume-weighted samples from the distributions of all incoming edges to the node. The sub-sample for residual sediment $\mathbb{R}_{residual}$ on the edge is drawn from the distribution for that edge $e^{\mathcal{N}_{edge}}$ at the start of the timestep; the sub-sample for sediment mobilised from the basal sediment layer $\mathbb{R}_{mobilised}$ is drawn from its distribution $e^{\mathcal{N}_{sediment}}$; finally, for sediment eroded from bedrock $\mathbb{R}_{bedrock}$ the sub-sample is drawn from the population distribution $e^{\mathcal{N}}$. From the union of these, $\mathbb{R}_{mixture}$, an updated $\mu$, $\varsigma$ and $d_{50}$ are defined for the edge. In the case of deposition to the basal sediment layer the new distribution for the basal sediment layer is defined as the volume-weighted union of the depositional component $\mathbb{R}_{deposition}$ sampled from the distribution for that edge at the start of the timestep and the residual component $\mathbb{R}_{sediment}$ sampled from the distribution for the basal sediment layer.

### 2.2.6 Detrital provenance tracking

In addition to the grain size, the network transport model can track mixtures of detrital properties, so yielding a provenance distribution at the outlet. Detritus tracking is entirely passive and can be omitted to save computational cost. Two modes are enabled in GraphSSeT; the 'normal' mode keeps track of the sediment source when sediment last joined the active transport system. Three source classes are defined: 'init' describes active sediment generated in the first timestep; 'basal' describes sediment mobilised from the basal sediment layer and 'bedrock' describes sediment derived from erosion that has never been deposited. These properties do not persist through cycles of deposition and remobilisation, hence sediment that has been eroded from bedrock, deposited and later remobilised will have 'basal' class.

Bedrock tracers such as isotopic data are important fingerprints of glacial erosion that can be reliably recovered from sediment cores (Licht and Hemming, 2017). Quantitative model-based approaches using these data can constrain cryopshere processes more reliably, but transport is a source of ambiguity in the detrital provenance problem (Aitken and Urosevic, 2021). Supporting this, the second 'bedrock' mode tracks a set of detrital source classes, for example, bedrock geology classes. In contrast to the 'normal' mode, these classes persist through sediment cycles and the characteristics of the basal sediment layer





also becomes defined by its constituent classes. In this mode, there is no distinction between multiply remobilised sediment and fresh bedrock erosion.

## 3 Numerical Implementation

Following from the model formulations of Werder et al. (2013) and Delaney et al. (2019), this work considers dynamically evolving R-channels as the predominant mode of channelised flow, and only channelised flow is considered to have sufficient flux to mobilise sediment and to transport it significant distances over the timescales of interest. Consequently, for a graph representation of the hydrology system we use the network of channels and their attributes derived from subglacial hydrology models. Sediment entrainment into the ice, englacial sediment transport and deposition via glacial-hydrological processes are not considered here; glacial transport could be defined to evolve in parallel to the subglacial fluvial transport using a multi-graph representation.

### 3.1 Graph representations of subglacial hydrology

GraphSSeT is based on representations of the channelised hydrology as a set of connected edges in a directed graph. In this work, all graphs are constructed and manipulated using the NetworkX module in Python (Hagberg et al., 2008). We build the graph directly from the GlaDS FEM mesh. From the FEM element edges several different types of edges are defined including perimeter edges, perimeter-contacting edges, and internal edges. Outlet edges are defined either from the FEM boundary conditions, if the flow reaches the model boundary as grounded ice, or where hydraulic potential is zero at either node, but not both, representing a virtual grounding line. Entirely floating edges, where hydraulic potential is zero at both nodes, are excluded from the graph. Directionality for each edge is derived from the hydraulic potential gradient direction. Edges are populated with properties required for the dynamic sediment model including spatial coordinates, edge type, edge length, edge direction flag, hydraulic potential gradient magnitude, channel sectional area, and channelised water flux.

Graph nodes contain the characteristics of the ice and distributed sheet flow. Several node types are defined, including floating nodes (hydraulic potential of zero), perimeter nodes, and predefined 'moulin' nodes where focused water input is to be applied. Head nodes are defined at nodes with no predecessor edges, outlet nodes are defined either at the downstream domain boundary or at the downstream node of outlet edges. Nodes are populated with required properties including spatial coordinates, node type, surface and bed elevation, hydraulic potential, effective pressure, basal ice velocity magnitude, and the thickness of basal water layer. An example of a graph representing a hydrological model scenario is shown in Figure 2.

Key to our approach is the definition, from this main graph, targeted subgraphs that enable a flexible and sparse representation of the channelised flow network and therefore allow more efficient computation, and variable return time to edges. Subgraphs are views of the main graph, and so define a hierarchical set of graphs that capture the most essential components of the network without sacrificing generality. For steady-state model runs, with no variation in the hydrology conditions, we define these subgraphs only once, however for non-steady-state model runs, in which the hydrology conditions vary, a fresh subgraph view is generated every timestep.



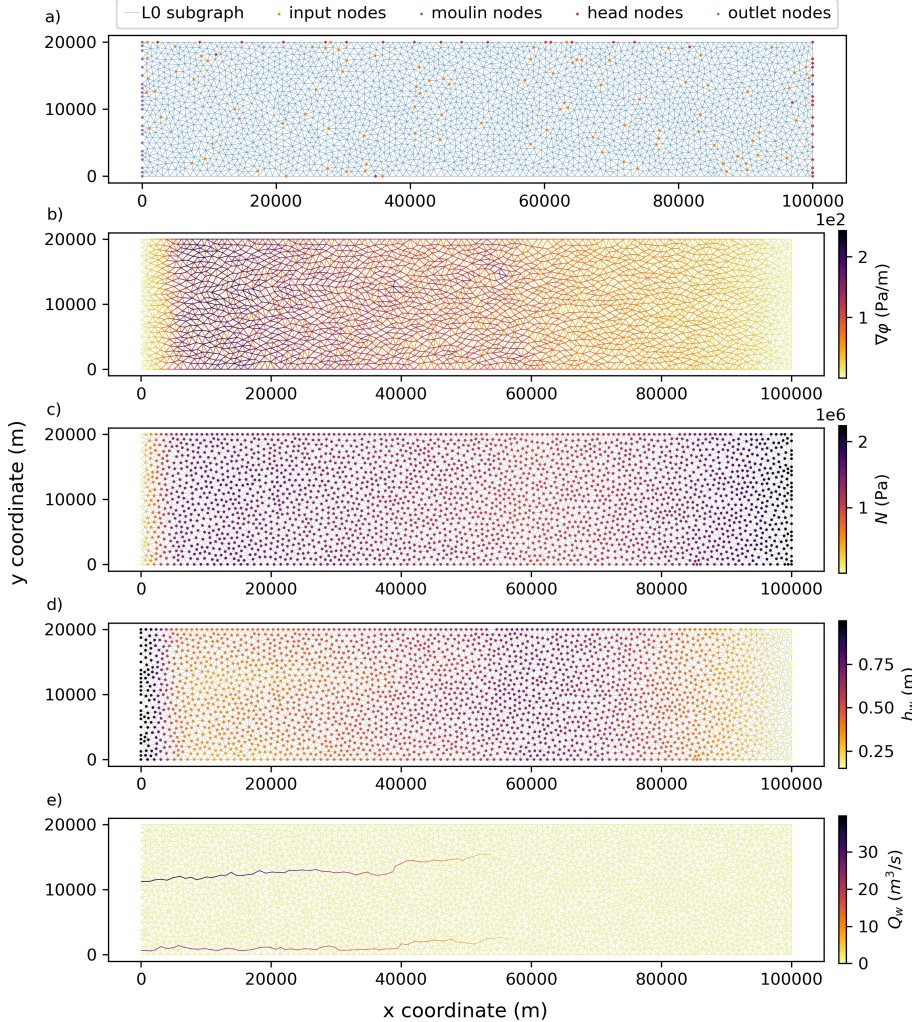

**Figure 2.** a) The level 0 graph representation of the SHMIP model scenario A5. Head nodes are nodes with no predecessor nodes, outlet nodes are defined at the 'grounding line'. Input nodes are a randomly selected subset of other nodes, used to define the network geometry. This model scenario has no moulin nodes. b) the hydraulic potential gradient defined on edges c) the effective pressure defined on nodes, d) the thickness of the sheet $h_w$ defined on nodes and e) the channelised water flux defined on edges

### 3.1.1 Graph connectivity measures for effective hydrology representation

320 Subgraphs may be defined according to several criteria, but our principal goal is to define for a given hydrology model scenario the most important edges to represent the channelised drainage system effectively, and so to reduce the magnitude and complexity of the model. Edge weights are assigned according to the channel area, which is a robust feature of the GlaDS model (equation 8) and is closely linked to the magnitude of water flux and the channel width. For the weighted graph we calculate edge betweenness centrality which represents the relative frequency that each edge is found on the shortest-paths between all





source node-target node pairs (Brandes, 2001). Source nodes include all head nodes and moulin nodes and a randomly selected set of $n$ input nodes representing the distributed hydrological inputs; target nodes are the outlet nodes.

From edge betweenness centrality we define hierarchical subgraphs as views of the main graph (level 0) for a more focused representation of the high-flux channels. In this case we define two further subgraph levels, with edge betweenness centrality $\geq$ 0 (level 1) and $\geq 0.005$ (level 2). For the model scenario A5, the level 1 subgraph includes 98% of total channelised flow on 31% of the edges, while the level 2 subgraph includes 97% of total channelised flow on 20% of the edges. Crucially, the high-flux edges are evolving dynamically with much higher sediment transport capacity and so we may analyse these subgraphs more frequently than the main graph, so reducing the computation cost without loss of precision. Besides computational benefit, the graphs for different model scenarios show a diversity of network characteristics that define how water (and sediment) are transported, with significant implications for understanding the sensitivity of sediment flow-organisation to changes in the hydrological system.

### 3.2 Sediment modelling approach

In line with section 2.2, we run the sediment transport model in several steps. For each model run we initialise the graph in the first timestep with required properties not defined by the input hydrology model scenario including initial grain-size distribution, $d_{50}$ and the basal sediment thickness as stochastic variables, grain density which is a constant here, and erosion potential. For each edge we calculate local transport capacity as in eq. 10 and sediment mobilisation as in eq. 13. The network-scale transport model is applied, and we update the bed elevation and basal sediment thickness as defined in eq. 15. The final, optional, stage of the GraphSSeT model is to apply detrital provenance tracking.

For the grain size distribution we use a stochastic sampling procedure. For steady-state model runs, the grain size distribution is seeded with a random number array that has, for each edge (or node) at each timestep a sample of size $n = 1000$ drawn from the standard normal distribution (mean = 0, standard deviation = 1). The input seed array spans the first two-weeks of the model time period, after which samples are drawn from a shuffled array. For non-steady-state model runs, every timestep we draw a new sample for every edge (or node) from the standard normal distribution. To generate the real valued distributions, each sample is scaled by $\varsigma$ and offset by $\mu$ for the relevant distributions.

## 4 Model scenarios and model inputs for this study

To define controls on the subglacial fluvial sediment transport system, a series of experiments were conducted with synthetic model scenarios derived from the Subglacial Hydrology Model Intercomparison Project, SHMIP (De Fleurian et al., 2018). We do not wish to study here the hydrology model differences and we model only examples from the 'mw' series of models computed with GlaDS.

Three series of SHMIP models were used. The A-series represent hydrologic steady-state with a distributed water input to the sheet ($m_b$) applied at a constant rate and distributed equally across all nodes in the domain. The B-series represents a hydrologic steady-state with both distributed and focused water inputs. Only a small distributed water input to the sheet





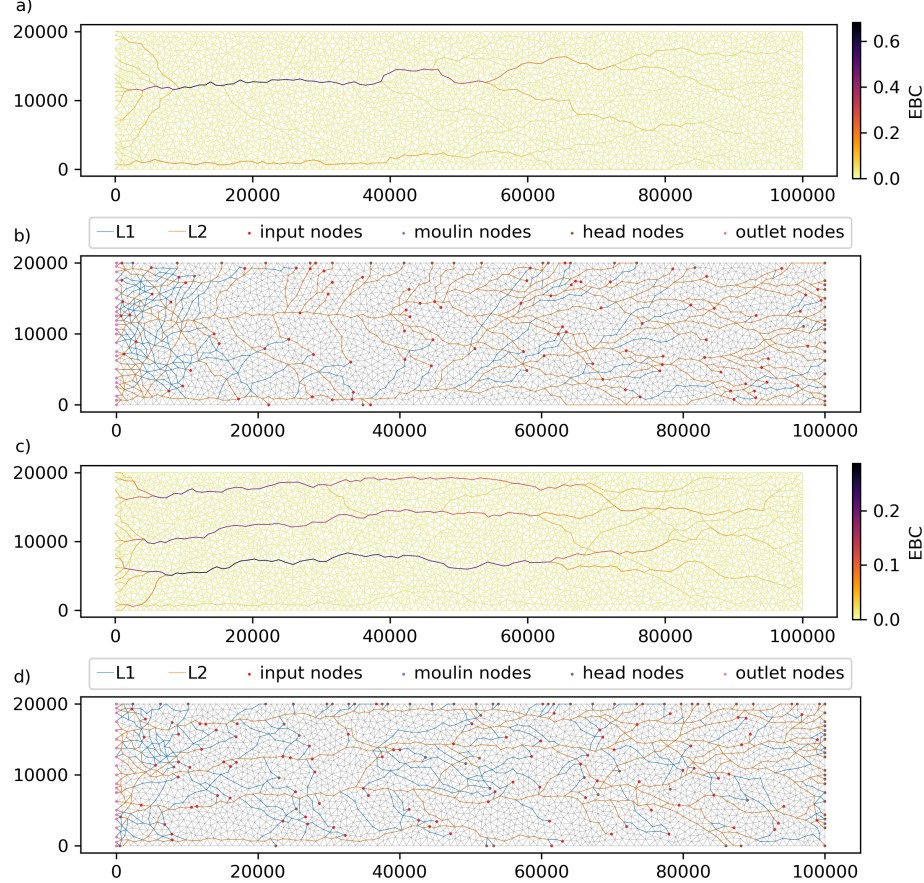

**Figure 3.** a) edge betweenness centrality (EBC) for the SHMIP model scenario A5 on the L0 graph, source nodes include moulins, head nodes and randomly selected input nodes, while target nodes are the outlet nodes b) L1 and L2 hierarchical subgraphs derived from EBC thresholds of $> 0$ and $\geq 0.005$ respectively. c) and d) show the same as a) and b) for model scenario A6. These examples do not have any moulin nodes

is included, while focused water input ($m_s$) is applied through moulin nodes, at a constant rate, but with differing intensity between scenarios. For these model scenarios, the hydrology network was developed dynamically over a time period of ca. 21 years. The C-series represents a hydrologic non-steady-state and involves time-varying water input to moulins on a diurnal cycle from $\frac{1}{4}$ to 2 times the base input. The C-series models build on the final state of the model scenario B5, with 100 moulins, and the hydrology network was developed over an additional time period of 50 days.

The base scenario for all experiments is the model scenario A5, with water input to the ice sheet bed at a moderate level ($4.5 \times 10^{-8}$ m/s or ca. 3.93 mm/day). Default experimental parameters and properties for our experiments are presented in table 1. For steady-state model scenarios the last timestep of the hydrology model scenario was used to force the sediment model. Most model runs were conducted over time period of 26 weeks, analogous to a summer season. The model run comprised daily cycles of 8 three-hour timesteps: the first and last use the level 0 subgraph, and in between a cycle of six timesteps alternating



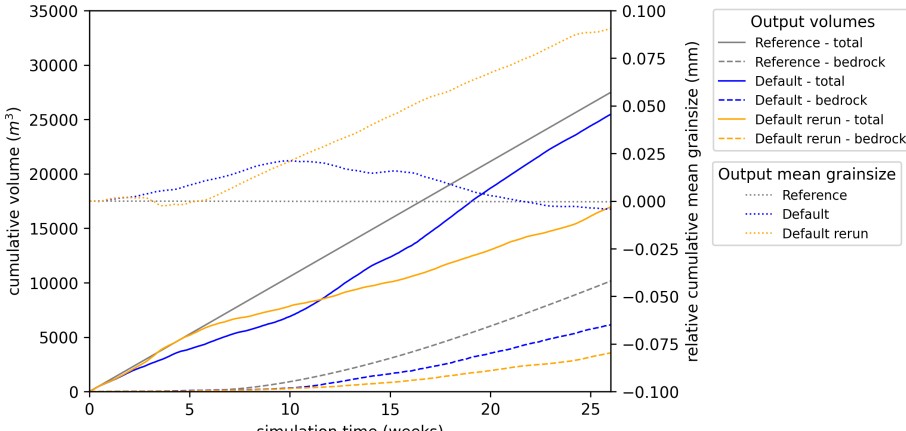

**Figure 4.** The reference run and two default runs for the hydrology model setup A5. Solid lines show the cumulative volume of sediment discharge, dashed lines show cumulative volume of sediment discharge from erosion of bedrock. Dotted lines show cumulative mean grainsize relative to the initial mean.

between the level 1 and level 2 subgraphs. This cycle captures the potential for higher-flux channels to evolve more rapidly, while ensuring the whole network is covered sufficiently often. For non steady-state model runs the sediment model is a downsampling of the original hydrology model scenario timesteps. This sampling need not be linear, but in this case, the hydrology

model state was reported every hour and we ran the sediment models at this sample interval.

Experiment set 1 was conducted for the A-series model scenarios with default parameters to demonstrate the effect of increasing basal water supply to the model. Experiment set 2 was conducted with the model scenario A5 and establishes the sensitivity to major parameters including the initial basal sediment layer thickness, the erosion scaling, $\Delta\sigma$, the grain size distribution, and grain density. Experiment set 3 considers the effect of focused water input using the B-series model

scenarios, while experiment set 4 considers diurnally time-varying water input using the C-series model scenarios. Finally, we demonstrate several model runs with the 'bedrock' detrital provenance tracking mode enabled using a graticule bedrock classification.

### 4.1 Reference and default models

For each hydrology model scenario, we ran at least two default model runs and a reference model scenario. The default model

has an initial randomly defined basal sediment thickness of $0.25 \pm 0.125$ m; and erosion scaling as $2.07 \times 10^{-7} u_b^{2.02}$ (Herman et al., 2018). The mean grain size is $\phi = 2.2 \approx 0.218$ mm and $\varsigma = 1.5$, corresponding to typical grain size distributions of subglacial till (Peterson et al., 2018; Haldorsen, 1981). The reference model scenario has the same variables as the default except for $\varsigma$ is set to zero, and so the outcome indicates the sediment model with no variation in grain size.

The hydrology model scenario A5 generates two major channels, one in the 'south' of the model and the other more central,

which has greater channel flux (Fig. 2e). Edge betweenness centrality subgraphs define a linear-dendritic network geometry



comprising numerous minor channels convergent with the major channels along their length and a divergent network near the outlet nodes. This divergent network accommodates the potential for flow to all outlet nodes (Fig. 3a). The reference model run generates a nearly constant sediment discharge rate at the outlet nodes (Fig. 4) representing the combined transport-capacity of the outlet edges. Total sediment discharge over 26 weeks for the reference model is $2.75 \times 10^4$ m$^3$, the vast majority from

the central subglacial channel. The default models show significant variations in sediment discharge rate occurring in line with the grain size distribution (Fig. 4). In these model runs the initial basal sediment layer is rapidly removed along the major channels, which subsequently remain largely sediment free, but more widespread mobilisation of sediment does not occur. The proportion of bedrock-derived sediment consistently increases from ca. 5 weeks on.

### 4.2 Experiment Set 1

In experiment set 1 we compare the impact of increasing basal water input for steady-state models run with default settings. Model scenario A4 considers a low-level of basal water input of 2.16 mm/day, and represents threshold conditions for chan-nelised flow. SHMIP model scenario A6 has a much greater input of 50 mm/day representing peak water discharge driven by surface melt in Greenland-like conditions (De Fleurian et al., 2018). Two additional GlaDS models were run in between A5 and A6 with flux rates of 21.6 mm/day (A7) and 39.3 mm/day (A8). The network geometry of all these models is linear-dendritic

with, as flux increases, the development of closer-spaced and longer channels (Fig. 5).

These model scenarios generate substantially different conditions for sediment transport with total sediment discharge for the default A4 model just $1.79 \times 10^3$ m$^3$, for A5 $2.55 \times 10^4$ m$^3$ and for A6 $4.60 \times 10^5$ m$^3$; the intermediate models A7 and A8 generate discharge of $1.01 \times 10^5$ m$^3$ and $3.72 \times 10^5$ m$^3$. Grain-size evolutions for the higher flux model runs show a consistent pattern of initially flat or increasing grain size for 5-10 weeks, followed by a systematic reduction in grain size.

These variations in grain size are associated with variations in sediment discharge (Fig. 6). The interpretation is that for higher flux model runs the selective transport of finer-grained material from upstream comes to increase discharge rates by increasing sediment transport capacity. Lower flux model runs show no such grain size reduction indicating that selective transport is not as significant in low-flux conditions.

### 4.3 Experiment Set 2

**4.3.1 Grain size distribution and its variance**

In experiment set 2 we vary selected input parameters (Table 1) from the default settings to gauge their importance for sediment transport. Grain size is fundamental to both transport capacity (equation 10) and transport velocity (equation 18) and also varies naturally across several orders of magnitude (Peterson et al., 2018; Haldorsen, 1981). Here we test the impact of this parameter, varying both $\mu$ and $\varsigma$ of the population distribution. In the former case (Fig. 7a) we see the profound impact of the value of $\mu$:

overall discharge for a mean of $\phi = 1.2$ (grain size 0.436 mm) is $7.96 \times 10^3$ m$^3$, but with a mean of $\phi = 3.2$ (grain size 0.109 mm) discharge is $6.25 \times 10^4$ m$^3$. Changes in the grain size distribution during the model run indicate that selective transport





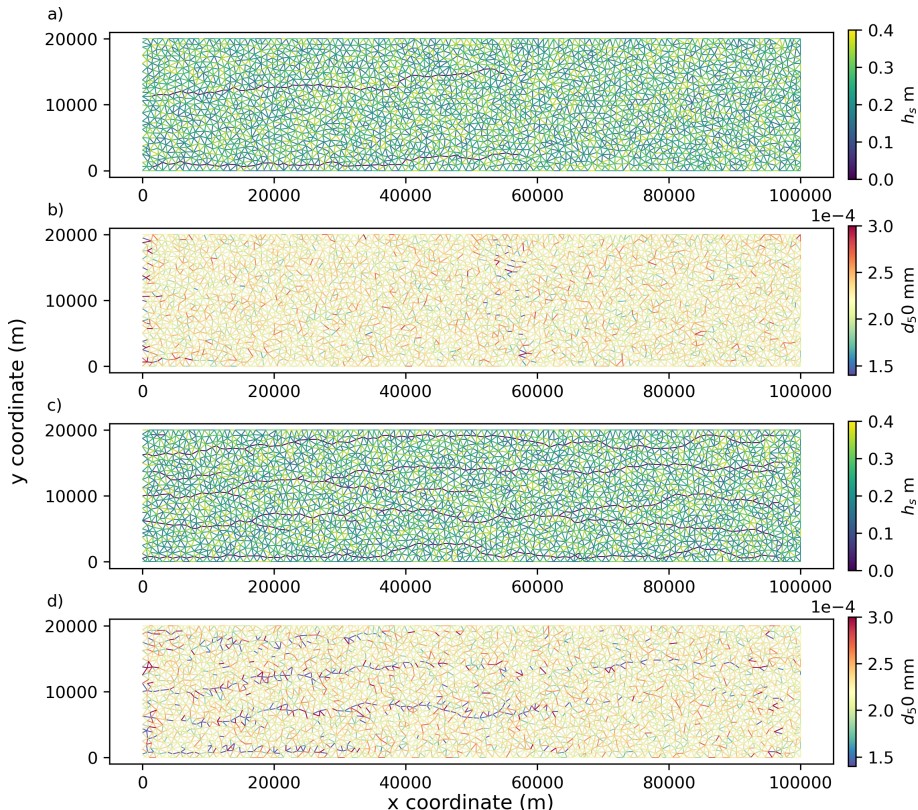

**Figure 5.** a) sedimentary layer thickness and b) median grain size at the end of the A5 default model run. c) and d) show the same for the A6 default model run. Note the selective transport for model A6 with finer-grained sediment in the major channels

**Table 1.** Tested parameters in Experiment Set 2, conducted with the model scenario A5

| Parameter | Default value | Test values |
|---|---|---|
| Mean grain size (mm) | 0.218 | 0.109; 0.436 |
| Grain size $\varsigma$ | 1.5 | 1;2 |
| Grain density (kg/m$^3$) | 2650 | 2550; 2750 |
| Initial sediment thickness (m) | $0.25 \pm 0.125$ | $0.05 \pm 0.025$; $0.5 \pm 0.25$ |
| $\Delta\sigma$ (m) | 0.001 | 0.005;0.01 |
| Erosion scaling | $2.7e^{-7}u_b^{2.02}$ | $1e^{-4}u_b; 2e^{-4}u_b; 1e^{-7}u_b^{2.02}$ |

occurs for the finer-grained model runs, causing a reduction in grain size from 10-15 weeks, however the coarser-grained model run does not show evidence of selective transport developing (Fig. 7a).

The variability of grain size is also significant with higher variance associated with increased discharge and finer at-outlet grain size, while lower variance is associated with reduced discharge and coarser at-outlet grain size (Fig. 7b). The interpre-



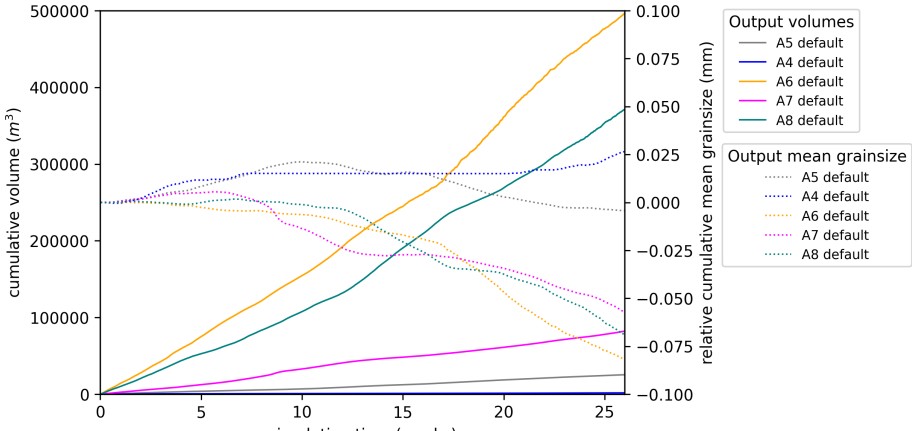

**Figure 6.** Sedimentary output for the models of experiment set 1. Solid lines show cumulative total sediment discharge and dotted lines show cumulative mean grain size relative to the initial mean.

tation is that samples with a coarser median grain size will reduce transport capacity and the associated sediment volume can be transported only slowly. In contrast, samples with finer median grain size, should be transported rapidly through the graph. Consequently, while higher variance will develop coarse and fine grained samples equally, only the fine-grained samples will propagate to the outlet. In Figure 7b with $\varsigma = 2$ strong selective transport reduces the grain size markedly from ca. 10 weeks

onwards, while for $\varsigma = 1$ the grain size increases steadily.

Grain density is in principle a very important factor for sediment transport capacity (eq 10), but due to a relatively limited range, its effect is minor in comparison to the grain size. The influence of a higher and lower grain density was tested, and showed that the model run with lower grain density is not associated with increased discharge overall, while the model run with higher grain density is associated with a moderately reduced discharge. This deviation may be explained by changes in

grain size, rather than necessarily a direct impact of grain density. Overall, grain density does not lead to large variations in either the sediment discharge volume or grain size at the outlet nodes.

### 4.3.2 Basal sedimentary layer thickness and $\Delta\sigma$

The thickness of the initial basal sediment layer is important for sediment transport because this material is available to be mobilised whenever transport capacity exceeds supply from erosion. We tested the impact of this initial condition with a

reduced initial thickness of $5 \pm 2.5$cm and an increased initial thickness of $50 \pm 25$cm (Figure 8). For these model runs, higher initial sediment thickness has increased total discharge (Figure 8). Although we may expect an enhanced supply for a greater initial thickness, the effect is in-line with the expected impact of grain-size variations. More significant are the delay in the onset of bedrock erosion to 15 weeks in the high initial thickness run, and a much lower proportion of bedrock-derived sediment (Figure 8).




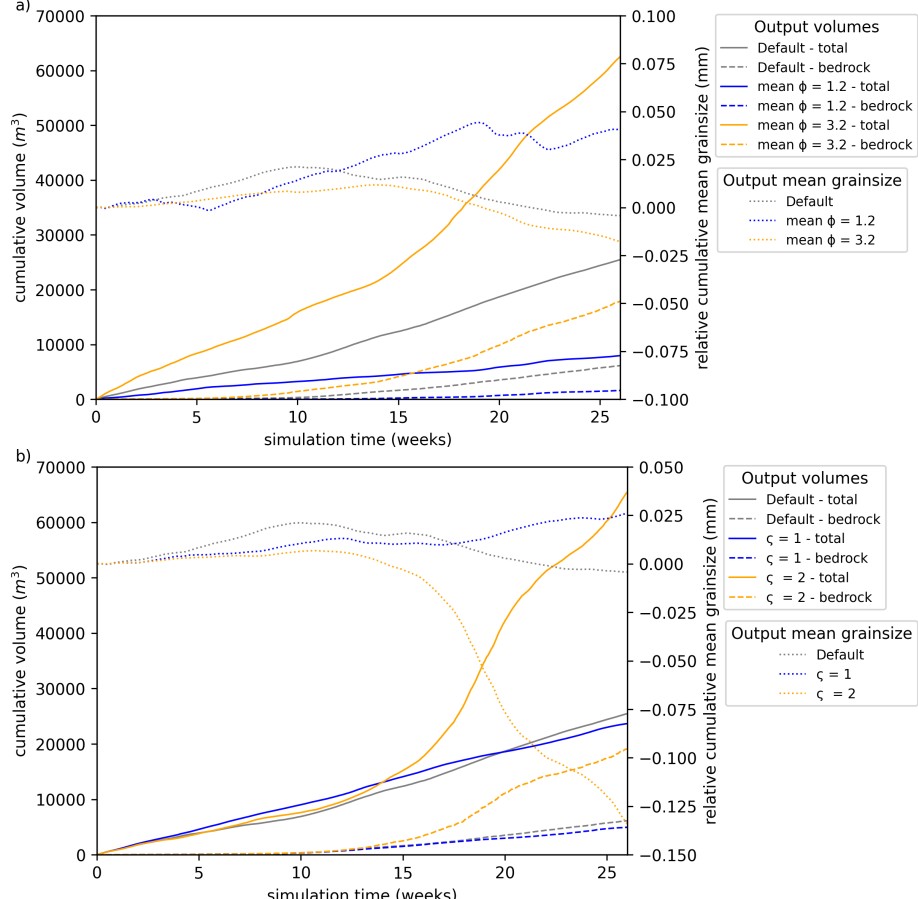

**Figure 7.** Sediment output with variations in a) grain size distribution mean and b) grain size distribution standard deviation. Solid lines show the cumulative sediment discharge volume, dashed lines show cumulative sediment volume derived from bedrock (never deposited) and dotted lines show cumulative mean grain-size relative to the initial mean

Sediment mobilisation is modulated by the parameter $\Delta\sigma$, which controls the transition between the supply-limited and transport-capacity limited regimes (eq. 13c). Increases to $\Delta\sigma = 0.005$ m and $\Delta\sigma = 0.01$ m caused little variation in the overall sediment discharge, but had a marked effect in reducing the proportion derived from the bedrock.

The importance of the basal sediment layer is not only increased supply, but also protection of the bedrock from erosion. $h_s$ and $\Delta\sigma$ control access to the bed and higher values for these sustain the basal sediment layer for longer. This enhances potential sediment supply and reduces the proportion of bedrock-derived sediment. Furthermore, in the case of deposition and remobilisation, a greater volume of basal sediment acts as a 'buffer' in grain size mixing calculations, which may influence grain size evolution and selective transport. Although important, the basal sediment layer had, for this model scenario, a small overall effect on the volume and grain size of sediment discharge.





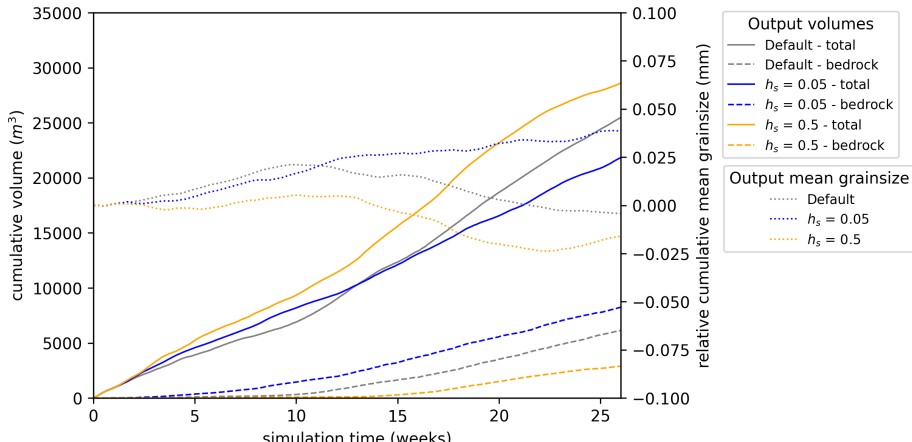

**Figure 8.** Sedimentary output with variable initial basal sediment layer thickness, $h_s$. Solid lines show the cumulative sediment volume, dashed lines show cumulative sediment volume derived from bedrock (never deposited) and dotted lines show cumulative mean grain-size relative to the initial mean

### 4.3.3 Erosion law scaling

The supply-limited regime, and its distinction from the transport-capacity limited regime, is fundamentally constrained by the rate of bedrock erosion (eq. 12), which in GraphSSeT is controlled first by the protective effect of the basal sediment layer, second by the basal ice velocity (constant at $1 \times 10^{-06}$ m/s in this case) and third by the choice of erosion scaling law. Velocity-based erosion scaling laws with linear and 'velocity-squared' laws are both common, although other exponents are possible (Herman et al., 2018, 2021; Cook et al., 2020). For our model scenario, with a 'velocity-squared' scaling, reducing the

pre-exponent had little effect on the model (Fig. 9). The linear scaling, in general, provided additional discharge and showed greater sensitivity to the scaling parameter (Fig. 9). However, the effect is much less marked than the variation in the erosion potential itself, which is a factor of ca. 30-60 times greater. The interpretation is that the effects of basal sediment thickness and capacity limited transport restricted the influence of erosion potential. More generally, if the system is transport capacity-limited and has an extensive basal sediment layer, then erosion potential will have a limited impact on sediment discharge. The

effect on supply is seen in the basement-derived sediment which has a greater volume proportion and earlier onset with the linear scaling law (Fig. 9).

### 4.4 Experiment Set 3

In experiment set 3 we investigate the influence of concentrated hydrological inputs with a scattered distribution (De Fleurian et al., 2018). In this experiment set the volumetric water input is identical to the A5 model scenario but comprises only a

small distributed basal input (0.006 mm/day), with the remainder split between $n$ randomly selected input nodes representing moulins. Models B1 through B5 investigate this flow for $n = 1$ to $n = 100$, with a corresponding volumetric water input for each of 90 m³/s to 0.9 m³/s. For each of these model scenarios we ran four sediment transport model runs. The results indicate



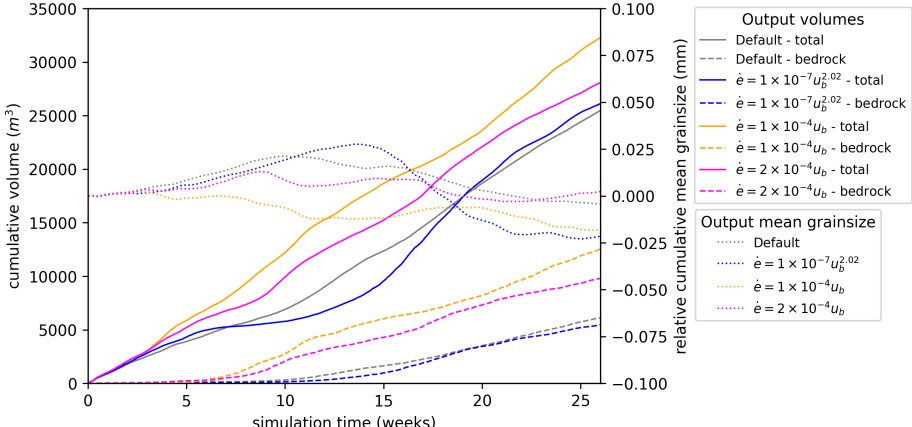

**Figure 9.** Sedimentary output with different erosion scaling laws. Scaling laws are for velocities expressed in m/a. Solid lines show the cumulative sediment volume, dashed lines show cumulative sediment volume derived from bedrock (never deposited) and dotted lines show cumulative mean grain-size relative to the initial mean.

that the distribution and intensity of focused water inputs are significant for sediment flux. The least sediment discharge occurs for the model scenario B1 with total discharge consistently below $5000 \, \mathrm{m^3}$. B1 has a single moulin that yields a single channel,

but is not effective in accessing the bed more broadly. The other model runs return variable sediment discharge, but in general the greatest discharge occurs for model B2, while B4 tends to have the least. These model runs have typically lower discharge than the runs for model scenario A5.

These model scenarios have only small difference in water flux at the outlet and in total channel volume, but they do have significant differences in the distribution and extents of channels and these are interpreted to dictate the sediment transport.

First, relative to model runs for scenario A5, the channel geometry is less well connected from the outlets to the inland areas of the domain, so limiting the area exposed to channelised flow. Second the main channels have typically reduced water flux. These combined effects lead to generally lower and more variable sediment discharge for a given total water input.

### 4.4.1 Experiment Set 4

For experiment set 4 we investigate the influence of non-steady state hydrology with a diurnal time-variable input over a period

of 50 days. The total volumetric input, and the locations of the moulins, are identical to the model scenario B5 but the input is varied according to a diurnal sinusoidal cycle

$$R(t, R_a) = max\left(0, m_s\left[1 - R_a \sin\left(\frac{2\pi t}{s_d}\right)\right]\right) \tag{26}$$

where $t$ is the time in seconds, $s_d$ the number of seconds per day and $m_s = 0.9 \, \mathrm{m^3/s}$. The amplification factor $R_a$ is tested across a range from 0.25 (C1) to 2 (C4), noting that for $R_a > 1$ it is necessary to truncate the function to remain non-negative

(eq. 26). This leads to a slight increase in total flux volume for the model scenario C4. We run in addition to these a C0 model which uses the same run procedure but without any diurnal variation.



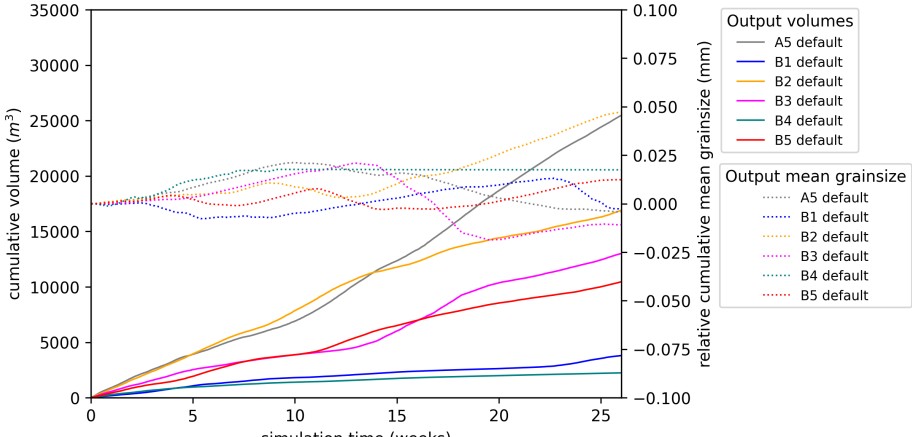

**Figure 10.** Sedimentary output for the default models of experiment set 3. Solid lines show cumulative total sediment discharge and dotted lines show cumulative mean grain-size relative to the initial mean. Basement components are not shown

For these runs, the sediment model is run with some modifications to the approach. With variable hydrology, edges may change status during the run (e.g. by becoming ungrounded) or flow-directions may be reversed. This demands that a fresh graph is constructed for each timestep, and from this graph we recompute edge betweenness centrality and generate the desired

subgraph. In this way the network geometry dynamically evolves with the hydrology model. We run the model with a one-hour timestep, in line with the hydrology model outputs and run in each day 8 cycles of three iterations, these being run on the L2, L1 and L0 graphs in that order.

The results (Fig. 11) indicate that the magnitude of the diurnal cycle impacts on sediment flux such that higher intensity in the diurnal cycle is associated with increased sediment discharge during periods of peak water input, while the corresponding

reduction during low-input periods is limited. The increased discharge is due to increased sediment concentration during daily peak flow, which reaches 60-100 ppm in the C4 runs versus peaks of 10-30 ppm for the C1 runs. The relationship is consistent with increased transport capacity at peak flow due to increased basal shear stress from fast water flow, in line with eq 10.

### 4.4.2    Detrital examples

For these model runs we track bedrock classes through the network. For our bedrock classes we use a graticule with three

classes across the model width and five classes from the front to the back (Figure 12a). Material eroded from nodes located within the graticule element are assigned the associated class for the purpose of detritus tracking.

The distribution of detritus for these model runs indicates significant differences in the erosion, mobilisation and transport of detritus. Systematic changes are seen in the degree of access to inland classes between models (Fig 12). For these model scenarios, erosion potential is uniform throughout and so the differences seen are due to bed access through sediment cover,

and the effectiveness of the transport system.



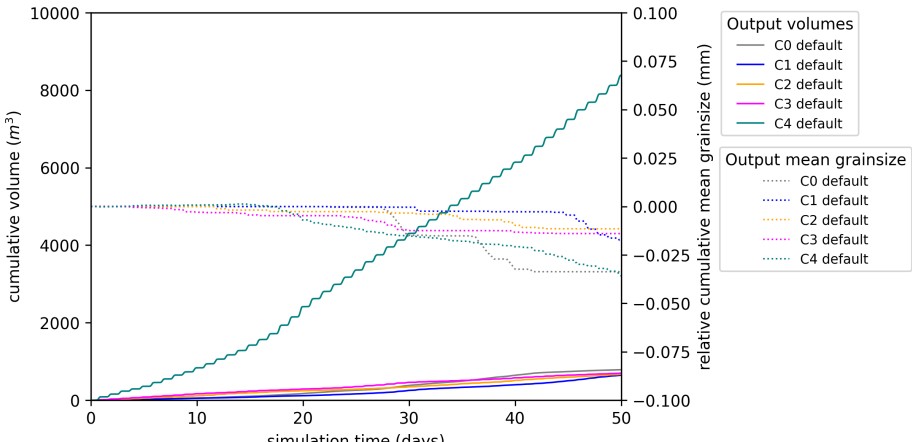

**Figure 11.** Sedimentary output for the models of experiment set 4. Solid lines show cumulative total sediment output and dotted lines show cumulative mean grain-size relative to the initial mean.

In all cases the early part of the model run is dominated by mobilisation of basal sediment, the volume of which reduces significantly during the model run as the initial basal sediment is depleted. For model run A4D the hydrology network is limited in inland extent and bedrock-derived detritus is dominated by classes '00', '01' and '02', reflecting transport at the margin, but with some class '11' due to the central channel. For model run A5D, the detrital signature is different, with the classes '22'

'11' and '01' dominant. These classes underlie the main channel where bedrock is exposed (Fig. 5). Moreover, the dominant class is '22' transported from the upstream catchment: as is expected from eq. 13, transport of incoming active sediment is prioritised over eroded or remobilised sediment and so the downstream classes '11' and '00' are subordinate to class '22'. High flux model runs A8D and A6D are markedly different to the preceding, with first a greater proportion of detritus from basal sediment, and second a much more uniform sampling of bedrock classes. This indicates a much broader sampling of the

bed, extending across almost the whole domain. Model run A6D has an especially enhanced contribution from class '20' from weeks 15 to 20. This corresponds to a period of low grain size, suggesting a strong grain-size driven selective transport event stemming from that region, propagating to the outlet.

## 5 Discussion

### 5.1 Geological controls on sediment flux

The dominant geological factor is grain size distribution, which controls transport capacity variations and in particular drives the evolution of selective transport. The transport system self-organises in response to stochastic variations in grain size: coarser grain sizes are not transported efficiently and tend to move slowly through the network; smaller grain sizes are preferentially transported and will tend to progress rapidly through the network. A tendency is seen in many of the model runs for increasing grain size early in the run (up to ca. 10 weeks) and then, for most higher-flux runs a systematic reduction in grain size. In





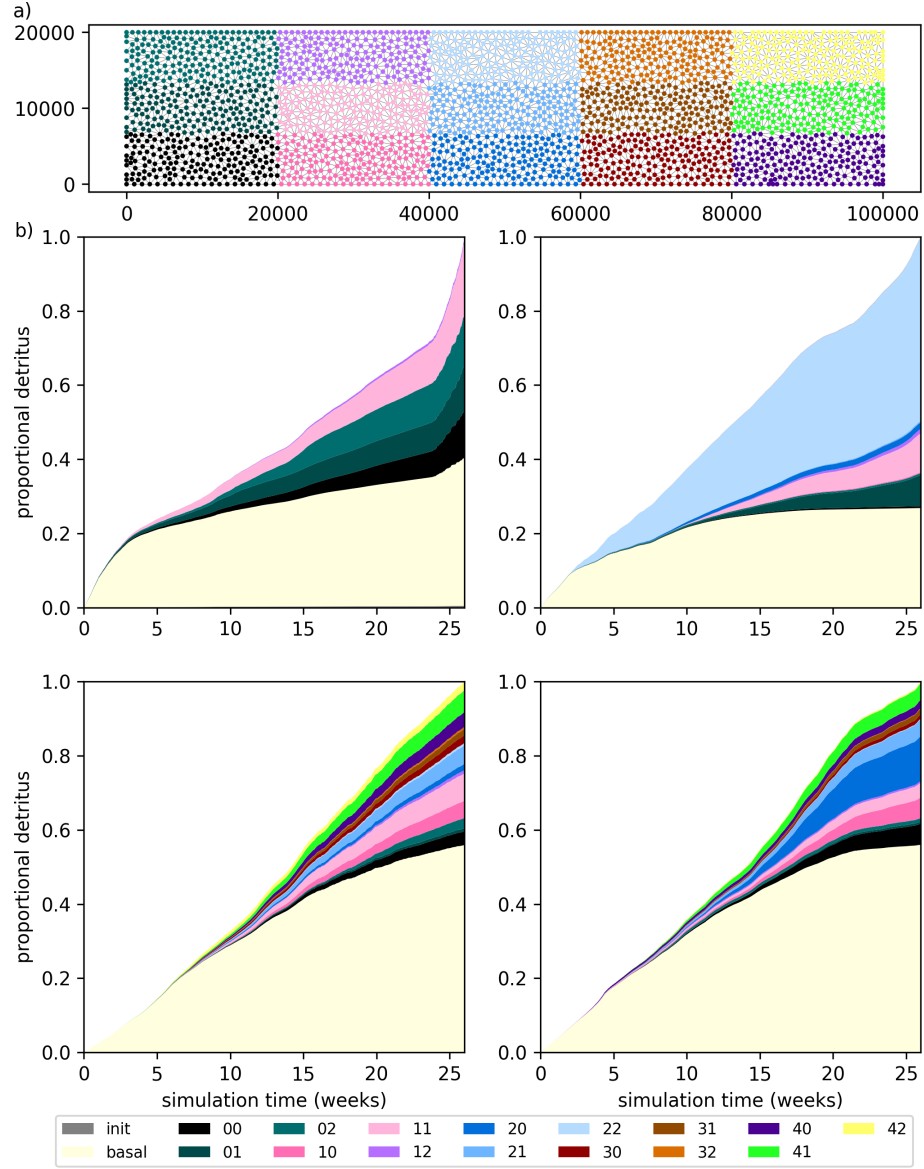

**Figure 12.** a) Map of graticule classes. b) to e) stackplots of cumulative volume by class as a proportion of total volume for the model runs A4D, A5D, A8D, and A6D. The class 'init' represents initial active sediment on edges, 'basal' represents the initial basal sediment layer; neither of these is associated with spatial location.

contrast most of the lower-flux model runs do not show any decrease in grain size. The evolution may be explained by the ice-sheet geometry, with a region of high potential gradient from 5 to 20 km upstream (Fig. 2) in which relatively coarse-grained sediment can be transported. The coarser-grained sediment can be transported off the network only slowly. In model runs with





higher flux, finer-grained sediment is transported from upstream due to selective transport, leading to a significant increase in transport capacity through time.

The thickness of the basal sediment layer is a key factor for sustained sediment supply and controls access to the bedrock (Delaney and Adhikari, 2020). A further effect of basal sediment thickness occurs during the mixing of sediment, as this layer is an important buffer for the evolving grain-size distribution and through this exerts further control on sediment transport dynamics. Particularly, the onset of volumetrically-abundant bedrock derived detritus is in many model runs coincident with significant reductions in the at-outlet grain-size distribution. This is a consequence of the feedback between transport capacity,

mobilisation, and grain size buffering. Under selective transport, sediment flowing into an edge will often have a finer grain size distribution than the residual sediment which will increase the transport capacity and is likely to cause remobilisation of basal sediment. In most cases this will be coarser-grained and so partially offset the increased transport capacity, but as basal sediment is lost the buffering effect of this layer is reduced and so the selectivity of the transport system is enhanced. For coarser-grained incoming sediment the reverse effect is expected and the drop in transport capacity will lead to deposition and

coarsening of the basal sediment leading to inhibited flow through the edge until sufficient finer-grained sediment arrives.

### 5.2   Glaciological controls on sediment flux

In our experiments, varying the erosion potential scaling law had a limited impact on sediment flux, as this factor is subordinate to the effects of the basal sediment layer, and because the model runs are transport-capacity limited. As a consequence, the composition of detritus reflects variations in transport rather than supply. This suggests that, for correct representation of the

detrital signatures of glaciers and ice sheets it is important to understand the transport as well as erosion (Aitken and Urosevic, 2021). Our input model scenarios have everywhere the same basal velocity, have no topography, and no variation in bed roughness or bedrock erosive properties. In the general case, variable erosion rates may occur due to variations in basal sliding velocity, in bedrock roughness and in erodability of the bedrock (Alley et al., 2019). A secondary effect is the potential impact of variations in bedrock lithology, joint frequency and structure orientation on grain size distribution (Krabbendam and Glasser,

2011; Hooyer et al., 2012). Variable erosion rates and grain size distributions would cause both sediment supply and transport capacity to vary in ways that might significantly impact network-scale transport dynamics.

### 5.3   Hydrological controls on sediment flux

#### 5.3.1   Water input controls on sediment flux

Experiment set 1 assessed the effect of basal water input on the discharge of sediment. For lower amounts of basal water input,

the channel system is less extensive but with higher amounts of basal water input these channels become more numerous and extend further inland. With no topography or other constraints in these model scenarios, the network geometry self-organises to form linear-dendritic channel networks, with little interaction between channels (Hewitt, 2011). In the model scenarios, the overall relation of sediment discharge to basal water input scaled linearly with the total channel volume. Across all model runs, the best fit scaling per unit width was $Q_s \approx 0.042 \sum Sl_{edge} - 1.28$ m$^3$/a. While low flux model scenarios have minimal





channelisation and low sediment discharge, higher-flux scenarios have extensive and well-formed hydrology networks. These networks have a much greater capacity to access the bed and to mobilise sediment across the broader domain (Figs. 5 and 12), leading to an increase in sediment discharge.

Experiment set 4 tested the effect of diurnal variations in water input via moulins. The model runs in this suite typically generated additional sediment flux relative to an identical series run without temporal variation. Furthermore, the greater the

variability the greater the sediment discharge, in particular for model C4 (Fig. 11). Due to a limited capacity for the channel geometry to adjust to short-term variations in water input, water velocity within the channels will increase, causing transient peaks and troughs in sediment transport capacity, and therefore, discharge. Furthermore, while the discharge peaks may be significantly higher than the background level, the corresponding troughs have little influence on total sediment discharge and so a systematically higher total sediment discharge is seen.

**5.3.2   Network geometry controls on sediment flux**

Experiment set 3 tested the effect of different configurations of moulins on the sediment flux, from broadly scattered inputs (100 moulins) to highly focused inputs (1 moulin), with the same overall flux distributed between them in each case. The single moulin model provided an exceptional result that one major high volume channel develops, but with little capacity to access the bed outside of this channel, and sediment discharge is always small. Sediment discharge for the other results were highly

varied, but were reduced relative to the model runs for the scenario A5. The configuration of moulin inputs is accompanied by significant changes in the network flow configuration, with less extensive channel distributions, and less well connected networks. The interpretation is that the extent and connectivity of channels is a limiting factor for efficient sediment transport at network scale. In the absence of other constraining factors, a completely distributed input will naturally self-organise with respect to forming regularly spaced channels and flow discrimination into 'catchments' within which flow-connectivity is high

(Hewitt, 2011), generating an efficient sediment transport network. With a network involving higher-flux at predefined input locations, network geometries must adjust to hydrological inputs that may not be well located with respect to the development of a channelised flow network (Gulley et al., 2012) and so the capacity to mobilise and transport sediment effectively at network scale is inhibited.

**5.4   Comparison with observational data**

We may compare our sediment discharge to estimates of total sediment flux from glacial systems around the margins of Greenland, for which estimates exist for Petermann Glacier of between 1080 and 1420 $m^3a^{-1}m^{-1}$ (Hogan et al., 2012) and Jakobshavn Isbrae of 1030-2300 $m^3a^{-1}m^{-1}$ (Hogan et al., 2011), and more generally estimates of sediment transport-rates in the range of hundreds to thousands of $m^3a^{-1}m^{-1}$. In contrast, total sediment discharge from our model runs is typically below 50 $m^3a^{-1}m^{-1}$, substantially less than observed volumes. We infer the low volumetric output of these model scenarios to be due to a

very small catchment size relative to natural systems, being only 2000 $km^2$ versus tens of thousands of $km^2$ for the catchments above.





Looking to sediment concentration, in the steady-state models the mean volumetric sediment concentration varied between 0.5 to 90 ppm, and was in the range ca. 20-30 ppm for most model runs. For a grain density of 2650 kg/m$^3$ these concentrations correspond to a range of ca. 1.3 to 240 mg/l, but typically in the range of 50 to 80 mg/l. Concentrations during high-flux periods were potentially much higher, often in the range of 200 to 400 ppm (or 500 to 1000 mg/l). Overall the sediment concentrations are within expected ranges for subglacial meltwater plumes reported in the literature (Chu et al., 2009; Schild et al., 2017; Overeem et al., 2017)

### 5.5 Model performance and development

#### 5.5.1 Limitations and further development

The model runs presented here have addressed the main drivers of subglacial fluvial sediment transport. The model runs do not cover long-term evolutions, which may involve new drivers that are superimposed on the above, for example, sustaining ongoing sediment supply becomes a more critical condition (Delaney and Adhikari, 2020) while seasonal to multiannual variations in hydrology superimpose additional variability to model forcing that may cause systematically different sediment transport conditions. Furthermore, we do not include topography, which is critical for hydrological flow organisation and will significantly control hydrology network geometry and flow-routing (Hiester et al., 2016). Variable ice sheet flow through space and time also will impact on channelisation and erosion, and may be a significant driver of variable sediment supply. The current model design is sufficient to address all the above scenarios with different input scenarios.

Grain size dependent transport has emerged as a key component of the transport model, with high sensitivity to changes in the grain size distribution causing selective transport and non-linear behavior. A more tightly-constrained grain size may in principle allow more consistent results, but this is unrealistic for glacial sediments that are generally unobserved due to ice cover, and in any case are characterised by highly variable grain size (Haldorsen, 1981). Alternatively, model ensembles are needed to mitigate the impact of the stochastic grain-size variations and to define with greater accuracy the expected sediment volume and characteristics for the glacial-hydrological scenario of interest. One limitation of the model presented here is that, although multimodal grain size distributions are likely to develop through mixing processes, these will be poorly represented by the unimodal grain size distributions used in this implementation. A Gaussian mixture model or a cumulative probability distribution approach could allow multi-modal distributions to be accommodated within the model design.

#### 5.5.2 Computational considerations

Model development has not yet emphasised computational performance, however some discussion is warranted. All model runs were conducted on a laptop with a 1.7 GHz processor (AMD Ryzen 7 PRO 4750U) and 32 Gb RAM, without parallelisation implemented. Run times were between ca. 1.5 and ca. 4.5 hours. Average timestep times for steady-state hydrology runs with 'normal' detritus tracking were ca. 4 seconds, increasing to ca. 4.5 seconds with 'bedrock' detritus tracking. For non-steady state models an average timestep took ca. 9 seconds, increasing to ca. 10.5 seconds with 'bedrock' detritus tracking.

The largest factor in run times was the size of the samples in the grain-size mixture calculations, for which a conservative choice of $n = 1000$ was used here to mitigate against excessive non-linear behaviour. A smaller sample size will yield signifi-
cantly reduced run times but with a greater variability in model outcomes. Significant potential exists for this performance to be improved upon through parallelisation and optimising the procedure for large-scale ensemble models run on high performance computing infrastructure.

## 6  Conclusions

We have developed a graph-based model, GraphSSeT, to represent subglacial hydrology networks and to calculate subglacial
fluvial sediment transport on those networks. The model uses the output of a subglacial hydrology model, for example GlaDS (Werder et al., 2013) as forcing, from which the channelised flow network is defined as a directed graph. The graph accommodates the definition of local sediment transport capacity and dynamic sediment evolution and the management of the flow of sediment at network-scale. GraphSSeT uses a stochastically-varying grain size which enables the evolution of the key process of selective transport. Grain size is tracked on the network as a distribution, and detrital properties may be tracked through the
network.

Using a set of synthetic model scenarios from SHMIP (De Fleurian et al., 2018), four experiment sets were run to investigate the impact of key factors in the model on sediment discharge volume, detrital characteristics and grain size. These include 1) the scale, degree of development and organisation of the subglacial hydrology network; 2) grain-size dependant transport generating an evolving selective transport and non-linear flow dynamics at network scale; 3) the effects of short-term variations
in water input for enhanced sediment output relative to steady state and 4) the evolving thickness of the basal sediment layer controlling sediment supply, access to the bedrock and buffering of grain size mixing processes. Overall, the results from these models generate sedimentary characteristics that are in-line with observations of sediment plumes and glacial sediments, although discharge volumes are very low compared to real examples, due to the small size of the model catchment.

*Code and data availability.* Code and model input data for these examples is on GitHub https://github.com/al8ken/GraphSSeT. Additional
input data are available at Zenodo (repository TBC)

## Appendix A:  Appendix A



**Table A1.** Summary of hydrology model runs referred to in this paper

| Input hydrology scenario | Total water input rate | Total channel volume | Outlet flux % channelised |
|:---:|:---:|:---:|:---:|
| - | $m^3a^{-1}m^{-1}$ | $km^3$ | - |
| A4 | 78840 | 0.349 | 81 |
| A5 | 143445 | 1.291 | 91 |
| A7 | 788400 | 9.540 | 100 |
| A8 | 1434450 | 17.426 | 100 |
| A6 | 1825000 | 21.795 | 100 |
| B1 | 142162 | 1.978 | 92 |
| B2 | 142162 | 0.906 | 91 |
| B3 | 142162 | 0.981 | 92 |
| B4 | 142162 | 0.841 | 92 |
| B5 | 142162 | 0.858 | 91 |
| C1 | 142162 | 0.893 | 92 |
| C2 | 142162 | 0.906 | 92 |
| C3 | 142162 | 0.912 | 92 |
| C4 | 173081 | 1.073 | 91 |



**Table A2.** Summary of Experiment Set 1 model runs. Bedrock-derived sediment values for 'D' models with bedrock detritus tracking are higher: these values indicate the total erosion-derived component regardless of sediment recycling

| Model run | Detritus mode | Total sediment | Bedrock-derived sediment | Mean grainsize | Mean sediment concentration |
|---|---|---|---|---|---|
| - | - | $m^3 a^{-1} m^{-1}$ | $m^3 a^{-1} m^{-1}$ | mm | ppm |
| A4 Reference | normal | 0.553 | 0.144 | 0.218 | 8.68 |
| A4 default | normal | 0.240 | 0.015 | 0.212 | 5.46 |
| A4 default_r | normal | 0.179 | 0.024 | 0.244 | 4.12 |
| A4D default | bedrock | 0.257 | 0.164 | 0.193 | 10.0 |
| A4D default_r | bedrock | 0.321 | 0.207 | 0.265 | 6.25 |
| A5 Reference | normal | 2.75 | 1.02 | 0.218 | 21.3 |
| A5 default | normal | 2.55 | 0.614 | 0.214 | 22.0 |
| A5 default_r | normal | 1.70 | 0.356 | 0.309 | 15.7 |
| A5D default | bedrock | 1.99 | 1.46 | 0.267 | 17.7 |
| A5D default_r | bedrock | 3.91 | 3.04 | 0.145 | 36.7 |
| A6 Reference | normal | 35.7 | 4.39 | 0.218 | 19.5 |
| A6 default | normal | 49.7 | 4.11 | 0.136 | 34.7 |
| A6 default_r | normal | 46.0 | 3.46 | 0.139 | 33.6 |
| A6D default | bedrock | 41.6 | 18.3 | 0.163 | 25.9 |
| A6D default_r | bedrock | 40.9 | 23.1 | 0.162 | 28.9 |
| A7 Reference | normal | 6.91 | 1.45 | 0.218 | 9.09 |
| A7 default | normal | 8.23 | 1.04 | 0.161 | 11.8 |
| A7 default_r | normal | 10.1 | 1.32 | 0.156 | 16.4 |
| A7D default | bedrock | 6.05 | 3.12 | 0.246 | 8.12 |
| A7D default_r | bedrock | 6.21 | 2.80 | 0.204 | 8.37 |
| A8 Reference | normal | 27.7 | 3.86 | 0.218 | 19.5 |
| A8 default | normal | 55.2 | 5.06 | 0.090 | 56.8 |
| A8 default_r | normal | 37.2 | 3.05 | 0.149 | 29.9 |
| A8D | bedrock | 41.8 | 17.7 | 0.123 | 34.4 |
| A8D_r | bedrock | 53.2 | 17.1 | 0.090 | 54.6 |



**Table A3.** Summary of Experiment Set 2 model runs, all with model scenario A5

| Variable | Value | Total sed. | Bedrock-derived sed. | Mean grainsize | Mean sediment conc. |
|---|---|---|---|---|---|
| - | - | $m^3 a^{-1} m^{-1}$ | $m^3 a^{-1} m^{-1}$ | mm | ppm |
| Mean grain size (mm) | 0.109 | 6.25 | 1.79 | 0.091 | 60.2 |
| Mean grain size (mm) | 0.436 | 0.80 | 0.16 | 0.477 | 7.67 |
| Grain size $\varsigma$ | 1 | 2.37 | 0.50 | 0.244 | 19.2 |
| Grain size $\varsigma$ | 2 | 6.55 | 1.91 | 0.084 | 92.2 |
| Grain density (kg/m$^3$) | 2550 | 2.63 | 0.56 | 0.239 | 21.7 |
| Grain density (kg/m$^3$) | 2750 | 4.13 | 0.89 | 0.124 | 37.9 |
| Initial sediment thickness (m) | $0.05 \pm 0.025$ | 2.18 | 0.83 | 0.257 | 17.9 |
| Initial sediment thickness (m) | $0.5 \pm 0.25$ | 2.86 | 0.29 | 0.202 | 24.6 |
| $\Delta\sigma$ (m) | 0.005 | 2.98 | 0.24 | 0.188 | 28.8 |
| $\Delta\sigma$ (m) | 0.01 | 2.39 | 0.096 | 0.239 | 19.8 |
| Erosion scaling | $1e^{-7} u_b^{2.02}$ | 2.61 | 0.55 | 0.196 | 28.2 |
| Erosion scaling | $1e^{-4} u_b$ | 3.23 | 1.25 | 0.200 | 27.2 |
| Erosion scaling | $2e^{-4} u_b$ | 2.81 | 0.98 | 0.220 | 24.3 |



**Table A4.** Summary of Experiment Set 3 model runs. Bedrock-derived sediment values for 'D' models indicate the total erosion-derived component regardless of sediment recycling

| Model run | Detritus mode | Total sediment | Bedrock-derived sediment | Mean grainsize | Mean sediment concentration |
|---|---|---|---|---|---|
| - | - | $m^3 a^{-1} m^{-1}$ | $m^3 a^{-1} m^{-1}$ | mm | ppm |
| B1 Reference | normal | 0.478 | 0.311 | 0.218 | 3.63 |
| B1 default | normal | 0.381 | 0.090 | 0.215 | 4.24 |
| B1 default_r | normal | 0.315 | 0.072 | 0.228 | 3.69 |
| B1D default | bedrock | 0.363 | 0.225 | 0.239 | 3.41 |
| B2 Reference | normal | 2.36 | 0.775 | 0.218 | 18.2 |
| B2 default | normal | 1.69 | 0.270 | 0.265 | 15.3 |
| B2 default_r | normal | 12.2 | 1.72 | 0.039 | 23.9 |
| B2D default | bedrock | 2.66 | 1.35 | 0.187 | 22.0 |
| B3 Reference | normal | 1.58 | 0.556 | 0.218 | 12.2 |
| B3 default | normal | 2.05 | 0.385 | 0.146 | 20.0 |
| B3 default_r | normal | 1.30 | 0.257 | 0.207 | 13.4 |
| B3D default | bedrock | 1.46 | 1.06 | 0.192 | 16.3 |
| B4 Reference | normal | 0.796 | 0.213 | 0.218 | 6.12 |
| B4 default | normal | 0.224 | 0.013 | 0.235 | 2.77 |
| B4 default_r | normal | 2.83 | 0.204 | 0.061 | 34.2 |
| B4D default | bedrock | 0.51 | 0.288 | 0.245 | 4.73 |
| B5 Reference | normal | 1.25 | 0.365 | 0.218 | 9.62 |
| B5 default | normal | 1.53 | 0.171 | 0.170 | 13.7 |
| B5 default_r | normal | 1.05 | 0.141 | 0.230 | 9.49 |
| B5D default | bedrock | 0.780 | 0.587 | 0.241 | 8.26 |



**Table A5.** Summary of Experiment Set 4 model runs. Bedrock-derived sediment values for 'D' models indicate the total erosion-derived component regardless of sediment recycling

| Model run | Detritus mode | Total sediment | Bedrock-derived sediment | Mean grainsize | Mean sediment concentration |
|---|---|---|---|---|---|
| - | - | $m^3a^{-1}m^{-1}$ | $m^3a^{-1}m^{-1}$ | mm | ppm |
| C0 Reference | normal | 0.091 | 0.004 | 0.218 | 0.700 |
| C0 default | normal | 0.288 | 0.003 | 0.184 | 3.02 |
| C0D default | bedrock | 0.196 | 0.096 | 0.196 | 2.76 |
| C1 Reference | normal | 0.098 | 0.003 | 0.218 | 0.833 |
| C1 default | normal | 0.237 | 0.003 | 0.201 | 2.96 |
| C1D default | bedrock | 0.947 | 0.054 | 0.038 | 11.2 |
| C2 Reference | normal | 0.173 | 0.004 | 0.218 | 2.29 |
| C2 default | normal | 0.245 | 0.041 | 0.206 | 3.24 |
| C2D default | bedrock | 0.240 | 0.143 | 0.208 | 3.28 |
| C3 Reference | normal | 0.275 | 0.023 | 0.218 | 4.21 |
| C3 default | normal | 0.255 | 0.005 | 0.204 | 4.31 |
| C3D default | bedrock | 0.269 | 0.169 | 0.208 | 4.27 |
| C4 Reference | normal | 2.74 | 0.129 | 0.218 | 37.1 |
| C4 default | normal | 3.06 | 0.145 | 0.182 | 42.2 |
| C4D default | bedrock | 3.66 | 1.63 | 0.166 | 48.8 |



*Author contributions.* Aitken ARA: Conceptualization, Methodology, Software, Investigation, Writing- Original draft, Writing - Review and Editing, Funding acquisition; Delaney IA: Methodology, Writing - Review and Editing; Pirot G: Methodology, Writing - Review and Editing; Werder M: Methodology, Writing - Review and Editing; Funding acquisition

*Competing interests.* One or more co-authors is a member of The Cryosphere editorial board.

*Acknowledgements.* This research was supported by the Australian Research Council Special Research Initiative, Australian Centre for Excellence in Antarctic Science (SR200100008). I. Delaney was supported by SNSF Project No. PZ00P2_202024. G. Pirot was supported by the Mineral Exploration Cooperative Research Centre whose activities are funded by the Australian Government's Cooperative Research Centre Programme.



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
