# Peer review of "Modelling subglacial fluvial sediment transport with a graph-based model, GraphSSeT"

_EGUsphere, 2024_

## Author Comment (AC2)

**Demonstration of edge-betweenness-centrality for imaging network flow structure**

[Figure]

The above shows a simple directed network. Flow on edges is directed from left to right and from the domain boundaries to the centre: flow cannot go right to left nor from the centre to the boundary. The graph may be thought of as several long channels, connected by shorter cross-channel segments. Several demonstrations follow that demonstrate the impact of network structure and weighting on the flow from the head nodes to the outlet nodes. In each, the upper image shows the weight (low weights are 'shorter' paths), while the lower image shows the EBC for flow from the head nodes to the outlet nodes.

Betweenness centrality of an edge $e$ is the sum of the fraction of all-pairs shortest paths that pass through $e$

$$c_B(e) = \sum_{s,t \in V} \frac{\sigma(s,t|e)}{\sigma(s,t)}$$

where $V$ is the set of nodes, $\sigma(s,t)$ is the number of shortest $(s,t)$-paths, and $\sigma(s,t|e)$ is the number of those paths passing through edge $e$. For the graph structure there are a total of 12 viable head to outlet node-pairs, 6 above the centre, and 6 below. For each node-pair, there are numerous shortest paths, potentially including multiple equal-weighted routes between these node-pairs.

Where the edge directionality and the weight parameter represent flow conditions, the EBC will represent the cumulative occurence of least-cost flow pathways for the sets of node-pairs provided, and therefore EBC-selected subgraphs will optimise for the most efficient flow pathways.

1) Equal weight

[Figure]

With equal weight, EBC reflects only the directional network structure. For both the top and bottom boundaries, EBC reduces down-flow from 3 at the head node to 1 at the outlet node as shortest paths move gradually to use the central line. On the central line, EBC increases slowly from 1 at the head node to (almost) 5 at the outlet node, where paths from all head nodes must converge. The intermediate lines have EBC of 2 because paths-in from the outer edges are balanced by paths-out to the central line.

2) 'Herringbone' weight

[Figure]

Increasing the across-flow weights relative to the down-flow weights shows a similar pattern to equal weight, but with a somewhat earlier transition to the central line as flow-orthogonal edges are abandoned in favour of diagonal edges.

3) Y-axis weighting

[Figure]

Applying a higher-weight based on Y-coordinate reduces the EBC of the entire upper boundary to 1, because shortest paths are redirected to the lower-weight central line as soon as possible. Similarly, in the lower portion, shortest paths are redirected to the lower boundary.

[Figure]

Reducing weighting towards the centre makes the central line the dominant route for shortest paths, with EBC of ~5 from 20 km onwards.

[Figure]

Reducing weighting towards the boundaries mitigates against inward migration of shortest paths which do not cross over to the central until required to after 180 km.

   4)   Random weighting

[Figure]

A random weighting leads to a more complex flow structure that is sensitive to details in the weightings. For example, on the lower boundary, a change in the weighting from low-weighted between 40-80 km to high weighted from 80-140 km, leads to a diversion of shortest-paths over to the intermediate line. In contrast, the upper segment has lower weights along the upper boundary and has EBC = 3 up until 180 km.

---

## Author Response (AR1)

Response to comments on EGUSPHERE-2024-274

**Modelling subglacial fluvial sediment transport with a graph-based model, GraphSSeT by Aitken et al.**

RC1- https://doi.org/10.5194/egusphere-2024-274-RC1

This paper introduced the GraphSSet model, a sediment transport model driven by subglacial channelized water inputs and producing rates of sediment output at the grounding line including grain size and volume, among other properties. It's really encouraging to see this work linking together catchment-level subglacial hydrology with sediment transport as it's an area that hasn't had much attention to date. I don't have any major comments but I've made a lot of targeted suggestions to clarify the writing and raise areas where it's less clear what the approach is.

*We appreciate the positive viewpoint of the reviewer and seek to improve clarity and clearness as detailed below.*

In general the model seems quite complex for application to subglacial settings where there are so many unknowns about the basal system. For example, knowing the sediment base thickness above bedrock would be very difficult to establish for most glaciers. It would be good to include a sentence or two in section 5.5.1 acknowledging how this would be approached for someone who would want to apply your model to a non-synthetic system.

*The point of GraphSSeT is to provide a flexible and versatile analytical environment for subglacial sediment transport. It is capable of complexity, which is needed to capture key processes, but for any application this may be scaled back as desired (e.g. a constant grain size, no detritus tracking, 'infinite' sediment layer etc). Best results are likely to be found through well targeted ensembles.*

*We have added a sentence as suggested indicating how complexity might be managed in real life studies … ongoing work is focusing on developing the approach for well-constrained models of real scenarios.*

In the same section, it should also be noted that because GlaDS is operating with R-channels rather than canals or channels eroded into sediment, further development of hydrology models is needed to better represent the interactions between sediment and subglacial drainage systems.

*GraphSSeT does not require any specific channel geometry but needs a geometry for which the basal shear stress from water flow can be calculated. Currently, a 'flat bedded' channel form is used that allows for R-channels and Hooke-channels, should these be the form used in the hydrology model. In principle, this could be extended to include rectangular, U-shaped and V-shaped channels fully or partially cutting into the sediment layer, should this become a feature of subglacial hydrology models. Integrated sediment-hydrology capability in GraphSSeT is currently rudimentary, but is a promising avenue for future improvement either within GraphSSeT or through coupling with hydrology models.*

*We make note here that, where a hydrology model is used as forcing, GraphSSeT needs to have a consistent channel geometry with the hydrology model. Therefore, in this case we use R-channels due to the R-channel geometry used in the input GlaDS models.*

Throughout the manuscript and for the figures you discuss the edge betweenness centrality. From the description of this as the frequency on the shortest path I don't understand how it can represent channels as suggested in the figures (I've not come across this term before). Is there a way to describe this for those not so familiar with graph models.

*We have provided a more extensive explanation of this concept in section 3.1.1. including now the base equation and its normalisation.*

*We note that EBC is not necessarily representing actual flow but the cumulative effect of network structure, input and output node selections and weights.*

*We attach to this response a demonstration of the concept on a simple directed graph – this I think does not easily fit into the main text, but could be an appendix or a supplementary item.*

I don't think I've ever seen a 94 page supplement! Can you include a guide at the beginning telling the reader what's included and the page range that those figures can be found.

*We feel it is necessary to include all experiment outputs. A table of contents is included with hyperlinks to the specific images so there is now no need to hunt through the PDF. Appendix tables are linked to the sections for each model run in the supplement.*

**Targeted line comments:**

Line 11: "grain size dependent selective transport" is a tricky combination of criteria to understand. It's the use of "selective" that I'm not sure about.

*The transport is selective in the context that fine grain sizes are transported more easily…hyphenation is added to make this more clear without making a lengthy sentence: "grain-size-dependent selective transport"*

Lines 13-16: I'm not sure what this means. Do you mean that more people are applying subglacial models?

*Indeed…reworded to "subglacial hydrology models are being applied more broadly and "*

Line 14 "quantitatively define"

*To define quantitatively is not incorrect…however we revise as suggested if it is more clear*

Line 15: "stronger connection" could be rephrased to say "enables constraining glacio-hydrological models with sediment records"

*Care must be taken here not to imply that a quantitative realisation of the sedimentary consequences of modelled hydrology is a 'constraint'. We rewrite as*

*'that, through comparison with the sediment record, may enable improved knowledge of the glacio-hydrological system and its impact on marine systems.      '*

Line 18: "impacts the delivery"

*revised as suggested*

Line 21: I don't think these are otherwise unobservable.

*Replaced with "difficult to observe in-situ"*

Line 23: "crucial for establishing"

*revised as suggested*

Line 24-25: This statement of subglacial evolution is vague.

*Clarified that we have a multiscale system spanning hours to centuries (at least)…but here we care mainly about the shorter timeframes.*

Line 24-30: These sentences are tricky to parse. Are you talking about Antarctic hydrology systems, Greenland or Alpine? There are lots of different drivers and timescales depending what systems you're examining. I'm not sure how these fit into the sediment discharge arguments from the sentences above. What about supraglacial lake drainage? I'm not sure how water piracy fits into the 'high flow' argument.

*We simplify and state this mainly in the general case – but for now excluding valley glaciers*

Line 32: To determine the consequences?

*Realize the consequences is better as we are not deterministic, but do make statistical realizations*

Line 45: Can you specify which inputs are distributed and which are point source inputs.

*Added "e.g. from basal melt" and "e.g. from input via moulins" respectively as being relevant to the examples here*

Line 46: Can you include a citation for water that might be stored englacially.

*We add a citation to Fountain et al (2005) for the concept and Werder et al (2013) which has the relevant formulation for GlaDS*

Line 62: "These include"

*Revised as suggested*

Line 66: 'coupled with ice flow' is confusing phrasing. In general 66-69 could be shortened given that you then say you don't focus on it.

*Phrase removed and this text shortened*

Line 67: 'move at the same speed as the ice'

*The speed of sediment motion in the deforming layer is not necessarily exactly that of the ice, so we avoid this phrasing.*

Line 70: why glaciated margins?

*As distinct from unglaciated margins … we simplified this sentence*

Line 93: 'basal water input' usually means water produced basally in situ which is not what I think you mean here.

*Revised to 'water input to the bed'*

Lines 85-112: most of this content isn't needed since you're using GlaDS. Since there are already a lot of summaries of glacial modeling approaches (e.g. Flowers at al, 2015) it would be better to reference those and reduce this section to a couple of sentences.

*See also comment of reviewer 2…we note that GraphSSeT is not depending on GlaDS, so other methods are relevant … therefore we retain reference to those, but in a shortened overview paragraph.*

Line 152: just saying you use pi is a bit misleading since S is a semicircle

*We revise this sentence "Here, $w_{c}$ is defined from the channel area $S$ using a Hooke angle of $\pi$ \citep{delaney_numerical_2019, hooke_subglacial_1990}, consistent with the semi-circular R-channel geometry in GlaDS."*

Line 203: I think point 1 needs to be rephrased to illustrate this is grain size dependent (assuming I've understood correctly) rather than just preventing excessive velocities for all sediment sizes

*We expanded all three criteria to make them more explicit. Specifically the virtual velocity is the expected 'particle velocity' for the median grain size*

Line 287: I don't understand what you mean by 'multiple remobilised sediment'

*We change to "In contrast to the `normal' mode, these classes persist through sediment cycles and the characteristics of the basal sediment layer also becomes defined by its constituent classes. In this mode, the detrital class is persistent through mobilisation and deposition, hence sediment that has been eroded from bedrock, deposited and later remobilised will retain its original class."*

Line 302: what do you mean by 'flow reaches the model boundary as grounded ice'. Wouldn't grounded boundary ice be defined in the FEM boundary conditions?

*We rephrase to be clear about the two conditions – "where ice becomes ungrounded (hydraulic potential is zero at either node, but not both), so representing a virtual grounding line, or where edges reach the downstream model boundary"*

*In the synthetic models the boundary condition is that the ice is ungrounded at the model boundary…so in this case it does not matter which*

Line 303: Could 'lakes' in the FEM model have hydraulic potential of zero?

*Yes, they do. In a model with lakes, their bounding nodes are considered 'outlets' … any incoming sediment will leave the model (sediment cannot be transported across the lake). There are no lakes (or sinks) in the models presented, however, as they are common in real life, we briefly comment on the handling of lakes and hydraulic sinks here.*

Line 307: if you're only using channel characteristics to drive GraphSSeT, why do you need information on the distributed system?

*Indeed, it is not needed for the algorithm but we carry this and other parameters into the graph for interpretation and visualisation…we removed the word required.*

Line 313: What does 'Key to our approach is the definition, from this main graph, targeted subgraphs' mean?

*Rephrased as "Our approach includes the definition, from this main graph, of flow-defined subgraphs that enable a flexible and sparse representation of the channelised flow network"*

Line 325: When you say the source nodes are a randomly selected set of input nodes there needs to be much more clarification. How can this represented distributed inputs? How many of these nodes are chosen?

*We add a clarifying paragraph as below:*

*To define transport on the graph it is necessary to have enough node pairs to represent the extent of valid flow paths. For maximal precision all nodes can be used, but to reduce computation time a sub-selection is used instead. Source nodes include all head nodes and moulin nodes and a random set of*

*$n$ input nodes to represent a spatially distributed hydrological input. Target nodes are the outlet nodes. In this case $n=100$ and with 37 head nodes and 24 outlet nodes, gives $\sim$ 3000 shortest paths. A directed graph will be less well sampled upstream relative to downstream.*

Line 329: you mention model scenario A5 but we don't know what this means yet.

*We add a reference here to section 4.2*

Line 379: you say you run at least two default models then only list parameters for one. I'm not sure 'reference' and 'default' are clear descriptions for these various runs.

*Yes, this is correct. The default model is run twice with the same parameters – the stochastic variation in grain size yields a different outcome every model run. The reference model has grain size variation of zero so is only run once.*

*We rephrase this sentence to make this more explicit.*

Line 395: specify whether this change is applied to GlaDS or GraphSSeT.

*The increase was applied in GlaDS as part of SHMIP – this is made explicit*

Line 437: the statement about the delay in the onset of bedrock erosion is interesting but I'm unsure how to read this in Figure 8. Is this from 'volume derived from bedrock (never deposited)'? Is it the yellow dashed line you're referring to? Very confusing here and in other figures what 'total' and 'bedrock' refer to.

*We clarify and simplify terminology used between the figure captions and the text*

Line 473: 'have only small differences'

*Revised as suggested*

Line 510: Can you direct readers towards Figure 12 again when you start talking about the classes.

*Revised as suggested*

Line 546: Do you mean "Our GlaDs input model scenarios have the same basal ice velocity, and no basal topography..."?

*We mean that there is no spatial variation of these variables in the model domain – reworded to express that more clearly.*

Line 644: At the end of your conclusion it would be good to include a statement of where the model can be applied to next e.g. applied rather than synthetic scenarios.

*We include a 'next steps' sentence highlighting the obvious applications to larger and more complex scenarios...*

Table A1: It would be helpful to include a column summarizing the main feature of the scenario e.g. moulins vs. distributed input (along the lines of Table 4 in DeFleurian et al, 2018)

*Included as suggested*

Figure 2a/3. Can you make the node dots larger in the legend – it's hard to see their color without zooming way in. Since you don't have any moulin nodes there's no point putting them in the legend.

*These dots have been enlarged in revised (new) figures for the paper…The EBC and L1/L2 networks shown now are not precisely those used in any model due to the random node set being different...It is the same from the point of view of defining the network geometry…*

*We prefer a consistent format across all figures so the 'moulins' stay for the legend but it is now obvious that these are not present - we add a comment in the caption to make this clear that they are not missing from the figure but absent from the model.*

*Figures in the supplement are not changed due to the need to recompute all results to do so*

Figure 4: I don't understand what 'total' and 'bedrock' refer to here or 'rerun'. This isn't described in the text.

*See comment above – we sort out the terminology in the captions make all this clear*

Figure 10. When you say 'basement' components do you mean bedrock? Consistent terminology would be good. I also don't see this figure listed in the text.

*Yes, it should be bedrock, revised and we checked the rest of the text for this error. Figure is now cited in section 4.4.*

**RC2- Citation: https://doi.org/10.5194/egusphere-2024-274-RC2**

This paper – "Modelling subglacial fluvial sediment transport with a graph-based model, GraphSSeT" by Aitken et al. – documents the development of 'GraphSSeT', which is a model capable of simulating the erosion, transport, and deposition of sedimentary material by hydrological systems beneath ice sheets. At this stage, the model is tested on 'synthetic' domains rather than real-world geometries. It enables the authors to make predictions about the nature of subglacial sediment transport and can be used to predict sediment characteristics such as grain size and volume, rates of sediment delivery, and sediment provenance information. The model is a complex and multi-faceted one, and the authors have been rigorous in exploring a wide range of parameter space. I have a few broader, overarching points, followed by a series of targeted suggestions that might help clarify the manuscript in specific places. Thank you for an interesting read.

*We appreciate the positive support for the model, which indeed is in the early stages of development.*

First, I would encourage the authors to be clear at the beginning of the methods section why such a complex model is required to address the problem. Section 2.2.1 goes into some quite specific detail of simpler hydrology models, but it is less apparent why these are insufficient for the problem at hand. What would be more useful is less of the specific detail here and more of a simple overview of the main limitations of existing hydrology models, which would help make the case for the development of GraphSSeT more clear.

*An important concept is that we do not model hydrology in GraphSSeT (at least not in this set of experiments) but use it to realise the sedimentary consequences of a hydrology model scenario. Any hydrology models discussed could be used **if** they have a definition of channels and some explicitly modelled or implied channel geometry (see comment to reviewer 1).*

*The complexity included in GraphSSeT itself is needed to resolve the observables desired (volume, grain size, detritus) and to represent the basal processes. Not all the complexity is needed for all examples - for example we have included in the code options to turn off variable grain size, the detritus tracking, or make sediment supply 'infinite' if we want only transport capacity.*

*At the beginning of section 2 we add a sentence to explain the need for a model that can do what GraphSSeT does…it is of course complementary to other approaches such as SUGSET that might be better for some applications.*

Second, and many of my specific comments pick up on this, the text needs clarification in several places to help the non-expert reader understand what the approach is, and why certain decisions have been made. The manuscript does appear to assume a lot of existing knowledge from the reader in places. A number of acronyms are not defined (including the name of the new model, which appears in the manuscript title!). I would also like to see more clarity on how this model (which is developed on a synthetic experimental basis) can be applied to 'real world' questions, perhaps with a couple of examples. I assume the authors are keen for the model / code to be used and build upon by the wider community, so providing a steer on suitable applications would be helpful.

*For the purposes of not muddling the issue and lengthening the paper further with extensive description of the real-world scenarios – which naturally are more complex than the synthetic models here - we prefer not to include a real example. Once a hydrology model has been converted to a valid graph representation, the process of application is the same, and the same codes can be applied to real. A parallel version is being developed to allow for large models.*

*We note in section 5.5.1 that GraphSSeT is being applied to 'real world' examples in Greenland and Antarctica…*

*And we have tried to make the text as lean and straightforward as possible*

Third, the supplement contains 167 figures! I appreciate that there is a lot of parameter space to be explored and the rigour in producing all these figures is commendable! I would make a few suggestions to help the reader parse this level of detail: (i) ensure relevant parts of the supplement are referred to in appropriate places in the main manuscript (I could find very little evidence of this), (ii) consider whether any figures could be grouped into some form of animation, or whether simply making the data behind the figures (and scripts used to generate them) available would suffice, (iii) include at the start of the supplement a guide or table of contents that helps the reader locate specific figures of interest.

*We include a table of contents with hyperlinks and refer to the relevant supplement sections in Tables A1 through A5.*

*A range of animations, data and scripts are made available in the Zenodo repository to enable full investigation of specific scenarios. But to regenerate all the figures would necessitate model reruns – which I doubt many would have time (or need) for!*

**Specific comments:**
Line 23: 'pinning points' is usually associated with a specific concept in glaciology, and I think its usage here could be misleading. Do the authors mean 'turning points'?

*Phrase is deleted*

Line 33: comma needed after 'cryosphere'.

*Not changed – the meaning is as we intended I think*

Line 62: Typo; should read 'These include'.

*Revised as suggested*

Line 66: This sentence didn't quite make sense to me (particularly 'sediment coupled with ice flow including as sediment…'). Consider simplifying (since this is not the focus on the study in any case).

*Simplified – see also reviewer 1 comment*

Line 74: suggest a full-stop after 'observations'. i.e. '…constrained by observations. However, they share…'.

*Revised as suggested*

Line 80: could you explain what the 'SUGSET model' is? Is this an acronym? I know more description comes in section 2.2.3, but spelling out where the name comes from (if indeed it comes from anywhere!) would be helpful.

*We rewrite this sentence to explain the basis of the SUGSET model. SUGSET is SUb-Glacial SEdiment Transport which is expressed here.*

Section 2.2.1: I think there is an unnecessary amount of detail here given that, if I understand correctly, these models are not used in this study. I think this section could be summarised in just a single, concise paragraph, particularly highlighting the shortcomings of existing models.

*Revised to a shorter statement– we note that although we use it here GraphSSeT does not depend on GlaDS to work. In particular, the network-based model of Schoof may be the most easily integrated with GraphSSeT…*

Line 112: Again, for those unfamiliar, what does 'GlaDS' stand for?

*GlaDS is the Glacier Drainage System model. The GlaDS acronym is explained here*

Line 141: As above, could you explain where the name 'GraphSSeT' comes from?

*It is a pronounceable portmanteau of Graph and SUGSET …. we provide a backronym here.*

Line 152: if section 2.2.1 is shortened, it may be worth clarifying at this point what the Hooke angle is. It isn't immediately apparent from reading the current text why pi is chosen as this angle (is it because of the semi-circular shape of the channel?).

*Rephrased (see comment from reviewer 1)*

Line 169: is 0.75 m simply chosen as a value between the typical lower/upper bounds? Can you justify this more clearly (i.e., why not use 1 metre or 1.25 metres?)?

*This value describes the point where sediment cover completely protects the bed from erosion and gives a smooth rather than abrupt transition to that point. While the rationale for this parameter is sound, the basis for selecting a specific value is not so well developed – it should not be deeper than the expected depth of deformation in the till layer. Some limits to the observed depth of deformation in till are given in Evans et al. (2006) between ~20-30 cm to 2 m or so. We follow this and prior applications to derive the likely limits…0.75m follows Delaney et al (2019).*

*In practice, this parameter acts together with sediment thickness to control access to bedrock; in the context of the model outputs these may affect sediment make-up (grain size and detritus), and only in a supply limited case the volume. We have revised this sentence to convey some more clarity around the impact of varying these parameters.*

Line 188: any particular reason for a porosity of 0.3? Standard value for subglacial sediment?

*Value is typical of reported values for tills between 0.2 and 0.4 – we add also a reference to Evans et al (2006).*

*While GraphSSeT can accommodate porosity, this parameter only matters for converting between sediment volumes (grains only) and sediment thicknesses (including pores), and can be considered as zero if the height limit ($h_{lim}$) and erosion depth ($h_{max}$) are suitably adjusted (see Delaney et al., (2019) for discussion).*

Line 203: could you clarify what you mean by an 'excessive' velocity? I know this is outlined below, but it would be helpful if this initial sentence could be more specific.

*These criteria are rewritten more precisely*

Line 250: for those unfamiliar with Krumbein's phi scale, does 2.2 have a unit? Is this value in logarithmic space?

*It is a logarithmic scale so has no unit - the relation of phi to grain size in mm is clarified here for the unfamiliar.*

Line 275: 'detrital properties' is vague. Can you be more specific or give some examples?

*detrital properties, which represent the source(s) of sediment and its characteristics such as bedrock geology.*

Line 307: when using 'sheet flow', it may be worth specifying that this is water sheet flow as opposed to ice sheet flow.

*Revised as suggested*

Line 313: I didn't understand the sentence starting 'Key to our approach…'.

*Rewritten (see comment to reviewer 1)*

Figure 2a: For my eyes it is very difficult to distinguish between the different types of nodes from the coloured dots. Either the colours are too similar, or the dots are too small. The same goes for all figuresof this format (sorry!). If there are no moulin nodes, then that probably shouldn't be in the legend. Also, 'SHMIP model scenario A5' needs explaining and/or citation(s).

*We have redrafted the figures in the paper to make the nodes larger.*

*For the supplement we cannot regenerate the figures without re-running the models, and due to the stochastic nature of the model the results would then be different. So, we apologise for the small nodes…we did update the script to always have bigger nodes in the future!*

Line 324: edge betweenness centrality is an important concept in this paper but is not necessarily going to be intuitive or familiar to many readers. Could a simple schematic diagram / graph be used to help illustrate this term?

*See response above to reviewer 1*

Line 329: Model scenario A5 has not been defined up to this point.

*Citation to section 4 is included*

Line 330: should there be a reference to a relevant figure here?

*Citation to figure 3 is included*

Line 342: it isn't clear which types of provenance information the model is encoded to track.

*The model can (in principle) track any property…but the most likely are some sort of class linked to source location, such as bedrock geology.*

Line 352: what is the 'mw' series of models? Sorry if I have missed this.

*'mw' is a particular subset of the SHMIP model set … we remove this detail*

Line 360: I found the description of the A-, B-, and C-series models a bit confusing. Without any background in SHMIP, it is hard to know what these terms are referring to. Scenarios A5 and B5 are mentioned specifically, but it is not clear why these were chosen. Table A1 is useful but could be referred to more clearly in this section of the text, so readers know where to look to understand what these scenarios mean. It would also help if Table A1 contained a concise, descriptive summary of the key characteristic(s) of each input hydrology scenario.

*We do need to maintain consistency with SHMIP but we now include in table A1 the additional information from SHMIP.*

Line 376: what is a 'graticule bedrock classification'?

*We omit this detail here as it is clear in figure 12 and is explained later*

Figure 4: I'm not sure I understand what the different lines show (particularly 'total' and 'bedrock'). I couldn't see this explained in the text up to this point either.

*Terminology is clarified in caption*

Line 457: use 'approx.' or '~' rather than 'ca.' when describing a quantity that isn't a date or timespan (see also lines 593 and 594).

*Revised as suggested*

Line 507: it is not clear what the numbering convention for these different classes is, and what they mean. Figure 12 helps a little, but I was still left uncertain how to interpret this section of text.

*The classes are simply spatial regions of the same size.*

*The classes are string codes not numerical values --- that is we can track 'names' --- for better ease of use we reclassify as 'left', 'center', 'right' relative to ice flow, and a number for the distance from the outlet. (e.g. 0L is the bottom left, 4R is the top right).*

Line 601: 'long-term evolution'?

*Rewritten as 'do not include any long-term runs'*